# Adversarial Crowdsourcing Through Robust Rank-One Matrix Completion

**Qianqian Ma**
Boston University
maqq@bu.edu

**Alex Olshevsky**
Boston University
alexols@bu.edu

## Abstract

We consider the problem of reconstructing a rank-one matrix from a revealed subset of its entries when some of the revealed entries are corrupted with perturbations that are unknown and can be arbitrarily large. It is not known which revealed entries are corrupted. We propose a new algorithm combining alternating minimization with extreme-value filtering and provide sufficient and necessary conditions to recover the original rank-one matrix. In particular, we show that our proposed algorithm is optimal when the set of revealed entries is given by an Erdős-Rényi random graph.

These results are then applied to the problem of classification from crowdsourced data under the assumption that while the majority of the workers are governed by the standard single-coin David-Skene model (i.e., they output the correct answer with a certain probability), some of the workers can deviate arbitrarily from this model. In particular, the "adversarial" workers could even make decisions designed to make the algorithm output an incorrect answer. Extensive experimental results show our algorithm for this problem, based on rank-one matrix completion with perturbations, outperforms all other state-of-the-art methods in such an adversarial scenario.[1]

## 1 Introduction

Matrix completion [10] [9] [13] refers to the problem of recovering a low-rank matrix from a subset of its entries. A fundamental challenge in the study of matrix completion is that, in some applications, the revealed entries will be inaccurate or corrupted. When these perturbations can be arbitrarily large, we will refer to the problem as "robust matrix completion." In particular, the motivating application for this paper is estimation of worker reliability in crowdsourcing [30] [41] [21][19][45], where this issue appears if some workers deviate from their instructions.

The simplest and one of the most widely used crowdsourcing models is Dawid & Skene's (D&S) single coin model [6]. In the D&S model, the workers are assumed to make mistakes independently of other workers and with the same error probability for each task. It has previously been observed that optimal estimation in the D&S model requires estimation of worker reliabilities [1], which in turn can be framed as a rank-one matrix completion problem [30].

In the case when some of the workers are poorly described by the D&S model or even consciously acting to subvert the underlying algorithm, the problem is naturally framed as a robust rank-one matrix completion problem. The motivating scenario is either a platform such as Amazon Mechanical Turk where workers without a long record of reliability typically have cheaper rates, or estimation from online ratings on a platform such as Yelp, where there is strong incentive for business owners to write fake reviews.

We propose a new algorithm for robust rank-one matrix completion which, in at least one regime, is provably optimal. We then perform a computational study which compares our method to twelve state-of-the-art methods from the crowdsourcing literature on both synthetic and real world datasets (in the latter case, we introduce the corrupted workers ourselves), and show that our method strongly outperforms all of them in this adversarial setting.

**Notation and conventions:** $[n] = \{1, \cdots, n\}$; $|S|$ is the size of set $P$; $\lceil x \rceil$ is the smallest integer greater than $x$; $\lfloor x \rfloor$ is the largest integer smaller than $x$; $\|X\|_*$ is the nuclear norm of matrix $L$, i.e., the sum of the singular values of matrix $X$; $\mathbb{Z}_+$ is the set of positive integers; $\mathbb{Z}_{\geq i}$ is the set of integers which are greater than $i$; Given $S_1$, $S_2$, the reduction of $S_1$ by $S_2$ is denoted as $S_1 \backslash S_2 = \{i \in S_1 : i \notin S_2\}$; finally, $A(n) \approx B(n)$ means $A(n)/B(n) \rightarrow 1$ as $n \rightarrow \infty$.

## 2 Related Work

**Matrix Completion:** the standard approach to low-rank matrix completion [2][3] usually proceeds by nuclear norm minimization:

$$\min_L \quad \|L\|_*$$
$$\text{s.t.} \quad [L]_{ij} = [L_0]_{ij}, \quad \forall(i,j) \in \Omega, \tag{1}$$

where $\|L\|_*$ is the nuclear norm of matrix $L$, $\Omega$ is the set of locations of the observed entries, $L_0$ is the matrix to be recovered. Candès and Recht [2] proved that $L_0$ can be recovered with high probability via solving (1) if $L_0$ is incoherent and $\Omega$ is sampled uniformly at random. These are strong assumptions and many papers, including this one, have sought to relax them. A popular approach has been to focus on non-uniform sampling. In particular, Negahban et al. [31] relaxed the condition of uniform sampling to weighted entrywise sampling. Király et al. [20] considered deterministic sampling. Liu et al. proposed a new hypothesis called "isomeric condition" in [27], which is weaker than uniform sampling, and proved that the matrix $L_0$ can be recovered by a nonconvex approach under this condition.

Unlike these general methods, we consider rank-one matrix completion problem with arbitrary $\Omega$ as well as adversarial corruptions. We do not assume any kind of incoherence of the underlying matrix. Of course, the rank-one matrix completion problem for uncorrupted cases is trivial and can be solved by going through the revealed answers recursively. Nevertheless, of particular note to us is Gamarnik et al. [10] which studied how well an alternating minimization methods in the uncorrupted case; we will build on those results in this work. Very closely related to our work is the recent paper Fatahi et al. [9], which considered solving the robust rank-one matrix completion with perturbations by solving the optimization problem

$$\min_{\boldsymbol{u} \in \mathbb{R}_+^m, \boldsymbol{v} \in \mathbb{R}_+^n} \|\mathcal{P}_\Omega(X - \boldsymbol{u}\boldsymbol{v}^\top)\|_1 + R_\beta(\boldsymbol{u}, \boldsymbol{v}),$$

where $\Omega$ represents the set of observed entries and $R_\beta$ represents a regularization term. The approach in [9] can deal with asymmetric matrices as well as sparse unknown perturbations. However, their approach has strong assumption for the structure of the graph $\mathcal{G}(\Omega)$ and can only deal with sparse perturbations.

**Crowdsourcing:** For crowdsourcing problems [46] [39] [35], the D&S model has become a standard theoretical framework and has led to a flurry of research in recent years, e.g., [41][11][15] among many others. In the D&S model, the workers are assumed to make errors independently with each other and the error probabilities of the workers are task-independent. In the single-coin D&S model, each worker is further assumed to have the same accuracy on each task.

Ghosh et al. [11] considered a binary classification problem based on single coin D&S model, and a SVD-based algorithm was proposed. The underlying assumption was that the observation matrix (representing which workers give answers for which tasks) is dense. To relax this constraint, Dalvi et al. [5] proposed another SVD-based algorithm which allows the observation matrix to be sparser. Karger et al. [18] extended the single coin D&S model to multi-class labeling issues, and they proposed an iterative algorithm to solve it. Zhang et al. [41] developed a two-stage spectral method for multi-class crowdsourcing labeling problems based on general D&S model. In [30], the skill estimate of workers in single one-coin D&S model were formulated as a rank-one matrix completion problem, which is also the approach we will adopt in this paper. We mention that in Section 6, we will compare our proposed algorithm with these existing approaches from [11][5][18][41][30].

The adversarial version of the D&S model, where adversaries represent workers who deviate from the model, has previously been investigated in a number of papers. Raykar et al. [34] assumed the adversaries are workers who give answers randomly, and they proposed approach to eliminate such adversaries in general D&S model. Jagabathula [17] also considered the specific types of adversaries (e.g all the adversaries provide a label $+1$) and tried to detect these adversaries from normal workers and eliminate their impact to the final predictions. Kleindessner et al. [21] proposed an approach to deal with arbitrary and colluding adversaries in the David&Skene model. Overall, the algorithm in [21] can deal with cases when nearly half of the workers are adversaries. However, this theoretical guarantee requires the task assignment matrix to be full matrix or dense matrix. Our work considers the same problem as [21] but without assuming the task assignment matrix is dense.

## 3 Problem Setup and Formulation

Let $X = M + P \in \mathbb{R}^{m \times n}$, where $M = \boldsymbol{a}\boldsymbol{b}^\top$ is a positive rank-1 matrix with $\boldsymbol{a} = \begin{bmatrix} a_1 & a_2 & \cdots & a_m \end{bmatrix}^\top$, $\boldsymbol{b} = \begin{bmatrix} b_1 & b_2 & \cdots & b_n \end{bmatrix}^\top$. Let $X_\Omega$ be a subset of the entries of $X$, i.e.,

$$X_\Omega = \{X_{ij} = M_{ij} + P_{ij}| \quad \forall (i,j) \in \Omega\}, \tag{2}$$

where $\Omega \subset [m] \times [n]$. Our goal is to efficiently reconstruct $M$ given $X_\Omega$. The elements of the matrix $P$ can take on any value; if $P_{ij} \neq 0$ we will say that the $(i,j)$'th entry of $X$ is "corrupted." An entry that is not corrupted will be referred to as "normal." We will be considering the situation where a number of rows and columns of $X$ are completely corrupted, and our goal is to correctly recover the remaining rows and columns.

We begin with a sequence of definitions. The structure of the set of pairs $\Omega$ can be conveniently represented by an undirected bipartite graph $\mathcal{G}(\Omega)$ as follows:

**Definition 1.** $\mathcal{G}(\Omega)$ *is defined to be a bipartite graph with vertex partitions* $V_u := \{1, 2, \cdots, m\}$ *and* $V_v := \{1, \cdots, n\}$ *and includes the edge* $(i, j)$, *with* $i \in V_u, j \in V_v$, *if and only if* $(i, j) \in \Omega$.

For $i \in V_u$, we let $\Omega_i$ denotes the set of neighbors of node $i$, and similarly for $j \in V_v$, let $\Omega'_j$ denote the set of neighbors of node $j$ (the apostrophe will be useful to be able to tell at a glance whether a node belongs to $V_u$ or $V_v$). Our next step is to formalize the fault model, i.e., what we will be assuming about the corruption matrix $P$.

**Definition 2.** *A set* $S \subset V(\mathcal{G})$ *is* **F-local** *if its intersection with each* $\Omega_i$ *and each* $\Omega'_j$ *has at most $F$ nodes.*

**Definition 3.** *A node* $i \in V_u$ *is* **corrupted** *if* $P_{ij} \neq 0$ *for some* $(i, j) \in \Omega$. *Likewise, a node* $j \in V_v$ *is* **corrupted** *if* $P_{ij} \neq 0$, *for some* $(i, j) \in \Omega$. *The graph* $\mathcal{G}(\Omega)$ *is said to be* **F-local corrupted** *if the set of corrupted nodes is F-local.*

Next, we will introduce some concepts from [25] dealing with the redundancy of edges between subsets of nodes; later, these will turn out to be closely related to the robustness of the graph $\mathcal{G}(\Omega)$ to corruptions.

**Definition 4.** *A set* $S \subset V(\mathcal{G})$ *is an* **r-reachable set** *if there exists a node in $S$ with at least $r$ neighbors outside $S$.* $\mathcal{G}(\Omega)$ *is* **r-robust** *if for every pair of nonempty, disjoint subsets of* $V(\mathcal{G})$, *at least one of the subsets is r-reachable.*

**Crowdsourcing:** we now explain the connection between robust rank-one matrix completion and crowdsourcing. We consider the single-coin D&S model where $W$ workers are asked to provide labels for a series of $M$-class classification tasks. The ground truths $g_t$ ($t \in [T]$) for these tasks are unknown (here $T$ is the number of tasks). The set $A \subset [W] \times [T]$ is a worker-task assignment set. The observations $(Y_{w,t})_{(i,t) \in A}$ are a collection of independent random variables. The single-coin D&S model supposes the accuracy of the worker $i$ is $p_i$ which means that the answer it returns is:

$$\mathbb{P}(Y_{i,t} = \ell | g_t) = p_i \mathbb{1}_{\{\ell = g_t\}} + \frac{1 - p_i}{M - 1} \mathbb{1}_{\{\ell \neq g_t\}}.$$

In words, worker $i$ returns the correct answer with probability $p_i$ and a random incorrect answer with probability $1 - p_i$. Under such assumptions, it was observed [30] that the probability of each worker can be estimated via solving a rank-one matrix completion problem as follows: letting

$s_i = \frac{M}{M-1}p_i - \frac{1}{M-1}$ we have that [30]

$$E\left[\frac{M}{M-1}\widetilde{C} - \frac{1}{M-1}\mathbf{1}\mathbf{1}^\top\right] = \boldsymbol{s}\boldsymbol{s}^\top, \tag{3}$$

where $\widetilde{C}$ is the covariance matrix between agents $i$ and $j$, $\mathbf{1}\mathbf{1}^\top$ is the all-ones matrix which has the same size as $\widetilde{C}$. Since the RHS of (3) is a rank-one matrix, the skill level vector $\boldsymbol{s}$ can be estimated by computing the empirical covariance matrix and applying a rank-one matrix completion method.

In the adversarial setting, we need to further consider the case when worker $i$ may deviate from the D&S model. In that case, all the entries in row $i$ and column $i$ should be viewed as corruptions as the derivation of Eq. (3) is no longer valid for that row; this is why we adopt a model in this paper where entire rows/columns are corrupted. Naturally, we do not know which rows/columns are corrupted. Extending our notation from above, we will refer to uncorrupted rows and columns as *normal*.

The hope is that identification of skill levels for the uncorrupted agents is still possible if the number of corrupted agents is not too large, or the corrupted agents are not placed in central location in the graph $\mathcal{G}(\Omega)$; making this intuition into a precise theorem is one of the goals of this paper.

With the above background in place, we can now state the main concerns of our paper formally:
(i) Given $X_\Omega$, how can we reconstruct the normal rows and columns of the rank-one matrix $M$ under $F$-local fault-models?
(ii) How can we estimate the workers' skill level $\boldsymbol{s}$ and consequently give accurate predictions for tasks in the single-coin D&S model with adversaries?

## 4 The M-MSR method

In this section, we present the details of our approach. We will start by explaining how our algorithm was constructed. For the uncorrupted rank-one matrix completion problem, we begin by observing that if $(\boldsymbol{a}, \boldsymbol{b})$ is an optimal solution, then we actually have the group of optimal solutions $(\boldsymbol{u}, \boldsymbol{v}) = (k\boldsymbol{a}, k^{-1}\boldsymbol{b})$. On the other hand, given arbitrary positive vectors $\boldsymbol{u}, \boldsymbol{v}$, we can represent them as

$$\boldsymbol{u} = \begin{bmatrix} a_1 k_1 & a_2 k_2 & \cdots & a_m k_m \end{bmatrix}^\top, \qquad \boldsymbol{v} = \begin{bmatrix} \frac{b_1}{k_1'} & \frac{b_2}{k_2'} & \cdots & \frac{b_n}{k_n'} \end{bmatrix}^\top,$$

where $a_i, b_j$ represent the values of optimal solution $(\boldsymbol{a}, \boldsymbol{b})$. We will refer to $k_i$ as the "value" of vertex $i$ and likewise for $k_j'$. Observe that to find the optimal solution, we need some algorithms to update $\boldsymbol{u}$ and $\boldsymbol{v}$ so that all the vertices can have the same value. To accomplish this, our starting point is the update

$$u_i(t+1) = \sum_{j \in \Omega_i} w_{ij} \frac{X_{ij}}{v_j(t)}, \quad \forall i \in [m],$$
$$v_j(t+1) = \sum_{i \in \Omega_j'} w_{ij}' \frac{X_{ij}}{u_i(t+1)}, \quad \forall j \in [n], \tag{4}$$

where the coefficients $w_{ij}$, $w_{ij}'$ form a convex combination. This update rule is motivated by [10], to which it is closely related, and can be interpreted in terms of an minimization method which alternates between finding the best $u$ and the best $v$. In the unperturbed case, via elementary algebra this can be rewritten in terms of the variables $k_i, k_j'$ introduced above as

$$k_i(t+1) = \sum_{j \in \Omega_i} w_{ij} k_j(t), \quad \forall i \in [m],$$
$$\frac{1}{k_j'(t+1)} = \sum_{i \in \Omega_j'} w_{ij}' \frac{1}{k_i(t+1)}, \quad \forall j \in [n]. \tag{5}$$

We will show with update rule (5), all the vertices can converge to the same value.

The main difficulty is what to do to account for corruptions: since the corrupted elements can be arbitrary, even a single corrupted element can completely destabilize this iteration. A natural approach is to filter the extreme values in each update of Eq. (4). To that end, let us define

$$J_i(t) = \left\{ \frac{X_{ij}}{v_j(t)} \middle| j \in \Omega_i \right\}.$$

We will then set $R_i(t)$ to be the set of nodes with the $F$ largest and smallest values in $J_i(t)$ (if there are fewer than $F$ values strictly smaller/larger than $u_i(t)$, then $R_i(t)$ contains the ones that are strictly smaller/larger than $u_i(t)$); the quantities $J'_j(t), R'_j(t)$ are defined similarly for nodes $j \in V_v$. Our algorithm is presented next; we will call it the Matrix-Mean-Subsequence-Reduced (M-MSR) algorithm.

---

**Algorithm 1** M-MSR

---

**Input:** Positive matrix $X$, set $\Omega$, $F$ and $\boldsymbol{v}(0) > 0$
**Output:** $\hat{X} = \boldsymbol{u}(T)\boldsymbol{v}(T)^\top$
1: **for** $t = 1, 2, \ldots, T$ **do**
2:　For each $i = 1, \cdots, m$, let

$$u_i(t+1) = \sum_{j \in \Omega_i \setminus R_i(t)} w_{ij} \frac{X_{ij}}{v_j(t)}, \tag{6}$$

　　where the coefficients $w_{ij}$ form a convex combination.
3:　For each $j = 1, \cdots, n$, let

$$v_j(t+1) = \sum_{i \in \Omega'_j \setminus R'_j(t)} w'_{ij} \frac{X_{ij}}{u_i(t+1)}, \tag{7}$$

　　where the coefficients $w'_{ij}$ form a convex combination.
4: **end for**
5: **return** $\boldsymbol{u}(t) = [u_1(t), u_2(t), \cdots, u_m(t)]^\top$, $\boldsymbol{v}(t) = [v_1(t), v_2(t), \cdots, v_n(t)]^\top$

---

For convenience, we will not consider the case $m = n = 1$. We will be assuming that $\mathcal{G}(\Omega)$ is connected. Finally, introducing the notation $\alpha$ for the smallest of $w_{ij}, w'_{ij}$, we will be assuming that $\alpha \leq 1/2$. This can easily satisfied by e.g., choosing $w_{ij} = 1/\text{degree}(i)$ and likewise for $w'_{ij}$.

## 5 Convergence Analysis

We begin by considering the case where the revealed entry set $\Omega$ is randomly chosen, which corresponds to the case of $\mathcal{G}(\Omega)$ being an Erdős-Rényi bipartite graphs. For simplicity, we assume $m = n$.

**Theorem 1(a).** Suppose $\mathcal{G}(\Omega)$ is a random bipartite graph $\mathcal{G}_{n,n,p}$ where each edge is generated with probability $p$. Then a sufficient condition for M-MSR algorithm (with parameter $F$) to successfully recover the normal rows and columns of $X$ under the assumption the the corruptions are $F$-local is

$$p \geq \frac{\log n + 2F \log \log n + x}{n}, \tag{8}$$

where $x = o(\log \log n) \to \infty$, when $n \to \infty$.

Thus, on a random graph, the M-MSR method can successfully recover the true matrix provided the graph is not too sparse. We will later discuss the guarantees this theorem implies on the total fraction of corruptions (note that $F$ is an upper bound on the number of corruptions in *each* neighborhood). For now we observe that Theorem 1(a) is tight in a sense described next.

As we will argue in the supplementary information, M-MSR has the property that it is skew nonamplifying in the following sense: at every step of the method, it maintains estimates $u(t), v(t)$ such that $u_i(t)v_j(t)$ converges to the correct answer $a_i b_j$ when $i, j$ are normal and which satisfy

$$\min_{\text{normal } i} \left\{ \frac{u_i(t)}{a_i}, \frac{b_i}{v_i(t)} \right\} \geq \min_{\text{normal } i} \left\{ \frac{u_i(0)}{a_i}, \frac{b_i}{v_i(0)} \right\}, \tag{9}$$

$$\max_{\text{normal } i} \left\{ \frac{u_i(t)}{a_i}, \frac{b_i}{v_i(t)} \right\} \leq \max_{\text{normal } i} \left\{ \frac{u_i(0)}{a_i}, \frac{b_i}{v_i(0)} \right\}, \tag{10}$$

Intuitively, the quantities $u_i(t)/a_i$ and $b_i/v_i(t)$ measure the "skew" between the vectors $u(t), v(t)$ and the optimal solution. Let us call any algorithm that satisfies Eq. (9) and Eq. (10) *skew-nonamplifying*.

Skew nonamplification is clearly a desirable property. In principle, any algorithm for robust rank-1 matrix factorization can converge to $(\alpha a)(\alpha^{-1}b^T)$ where $ab^T$ is the true rank-1 matrix and $\alpha$ can

be any real number. It is natural to bound how large the constants $\alpha$ and $\alpha^{-1}$ can be. The skew-nonamplifying property does that by ensuring the final skew is not worse than the skew on the initial conditions.

Our next result shows that M-MSR is essentially optimal on random graphs among all skew-nonamplifying methods.

**Theorem 1(b).** *Suppose $\mathcal{G}(\Omega)$ is a random bipartite graph $\mathcal{G}_{n,n,p}$ where each edge is generated with probability $p$. Suppose*

$$p = \frac{\log n + 2F \log \log n - x}{n},$$

*where $x = o(\log \log n) \to \infty$, when $n \to \infty$. Then, with probability approaching $1$ as $n \to \infty$, the normal rows and columns of $X$ cannot be recovered by any skew-nonamplifying algorithm in the presence of $F$-local corruptions.*

In other words, if Eq. (8) just barely fails due to the replacement of $x$ by $-x$, then Theorem 1(b) tells us that $\mathcal{G}_{n,n,p}$ will, with high probability, be a graph for which there exists a set of $F$-local corruptions which prevent any skew-nonamplifying algorithm from recovering the true matrix $X$.

Theorem 1 in parts (a) and (b) provides a justification for the M-MSR algorithm: it is essentially an optimal algorithm to use on bipartite random graphs. This theorem is actually derived from the following somewhat more general theorem, which gives the exact conditions for the M-MSR algorithm to work on an arbitrary graph.

**Theorem 2.** *Suppose $\mathcal{G}(\Omega)$ is a connected graph where the nodes are updated according to M-MSR algorithm with parameter $F$. Under $F$-local nodes-corrupted model, the rows and columns of $X$ without corruptions can be correctly recovered by the M-MSR method if and only if $\mathcal{G}(\Omega)$ is $2F + 1$-robust.*

Theorem 2 gives substance to the intuition that recovery is possible if the number of adversarial agents is not too large, and if their placement in the graph is not central. It quantifies this intuition through the concept of $2F + 1$ robustness. The proofs of these theorems can be found in the supplementary information. Our results are connected to earlier work in resilient consensus [25] which introduced the concept of robustness, as well as the work [40] which analyzed the threshold of robustness in general random graphs.

### 5.1 Applying M-MSR to crowdsourcing

We have already spelled out how skill determination in crowdsourcing with adversaries can be reduced to rank-one matrix completion with perturbations. Here we discuss additional details that are needed to apply the M-MSR method.

**Prediction.** When the skills are known, according to [26], the optimal prediction method under D&S model is weighted majority voting, i.e.

$$\gamma_{s,A}^*(Y) = \arg \max_{\ell \in [M]} \sum_{i:(i,t) \in A} v_i^* \mathbb{1}\{Y_{i,t} = \ell\}, \tag{11}$$

where $v_i^* = \log \frac{(M-1)p_i}{1-p_i}$, $\forall i \in [W]$. As discussed above, we will use perturbed rank-one matrix completion to estimate the skills.

**Implications of Theorem 1.** Consider the case where a total of $\beta n$ adversaries exist among $n$ workers, where $\beta < 1$. Of course, it is unknown who is an adversary. As explained earlier, a certain correlation matrix between the normal workers is rank-1 in expectation. Of course, we may not know all the entries of this correlation matrix, since only correlations among workers with tasks in common are revealed. A natural approach is to create a random bipartite graph of revealed entries by assigning tasks randomly.

This can be done in a number of ways. We may, for example, generate a random bipartite graph $G$ from $\mathcal{G}_{n,n,p}$ first. Then, assuming there is a sufficiently large incoming stream of tasks, we assign each task to a random pair of workers $i$ and $j$ such $(i, j)$ is an edge in $G$. After each pair of agents has been assigned enough tasks, the empirical correlation matrix is approximately rank-1, and we reveal the entries of this matrix corresponding to $G$. Note that, even though the correlation matrix is symmetric, this method will reveal an asymmetric subset of entries. Other methods to generate the

graph randomly via random task assignment are also possible, for example by assigning tasks to more than two workers. The key point, however, is that the fraction of adversaries in every neighborhood will then concentrate around $\beta$: for each node $i$, each of its randomly chosen neighbors is adversarial with probability $\beta$.

How many adversaries can we have and still correctly recover the skills of all the normal workers? Unfortunately, using the strategy of the previous paragraph, any constant fraction $\beta$ of adversaries will result in a failure. Indeed, glancing at Eq. (8), if $F$ scales as $\beta pn$ ($pn$ is the expected degree, and an expected fraction $\beta$ of these nodes will be adversarial), then Eq. (8) can never be satisfied.

Although this sounds discouraging, the guarantees of Theorem 1 are still useful, as we explain now. Glancing at Eq. (8), it is easy to see that the choice of $\beta = 1/[(2+\epsilon)\log(\log(n))]$ (and corresponding $F = \beta pn$) leads to that equation being satisfied with the choice of $p = \Omega((1+\epsilon^{-1})\log n)/n$. In other words, we can tolerate a fraction of $1/[(2+\epsilon)\log(\log(n))]$ of adversaries.

Fortunately, $1/\log(\log(n))$ decays to zero quite slowly. Recall that in the crowdsourcing scenario, $n$ will be the number of users; using an upper bound of 10 billion people for the population of planet Earth, we see that on any real-world data set, we have $1/[2\log(\log(10^{10}))] \approx 16\%$, which is a healthy proportion of adversaries to tolerate.

**Sign Determination.** In the M-MSR algorithm, we assume the rank-one matrix to recover is positive. However the rank-one matrix which we aim to recover in crowdsourcing problem is not necessarily positive. In fact, a worker's skill level $s_i \in [-\frac{1}{M-1}, 1]$ as $s_i = \frac{M}{M-1}p_i - \frac{1}{M-1}$, $p_i \in [0,1]$. To solve this issue, we can compute the entry-wise absolute value of the rank-one matrix, then apply M-MSR to get $|s|$, finally, we apply a post-processing step to identify the sign pattern of $s$. Details are available in Supplementary Sec. I.

# 6 Experiments

In several crowdsourcing experiments, we will compare the average prediction error $(\frac{1}{T}\sum_{t=1,\cdots,T}\{\hat{Y}_t \neq g_t\})$ of M-MSR algorithm with the straightforward majority voting (referred to as MV) and the following methods from the literature: [18](KOS), [11](Ghost-SVD), [5]( EoR), [41] ( MV-D&S and OPT-D&S), [30](PGD), [28](BP-twocoin, EM-twocoin, and MFA-twocoin), [42](Entropy(O)), [44] (Minmax) . A detailed description of all of these methods can be found in Supplementary Sec. A. In all cases we choose the corrupted workers at random, and the reported results are from an average of 50 runs, with the shaded region denoting the standard deviation.

**Synthetic Experiments.** Figure 1 shows the results of a number of experiments we have generated. We want to study the impact of graph types (Figure 1(b), 1(d)), task assignments (Figure 1(c), 1(f)) and skill level of "normal" workers (Figure 1(e)) to the performance of the crowdsourcing methods. Additionally, we want to study the impact of different adversarial strategies (Figure 1(g), 1(h), 1(i), 1(k), 1(j)). To do this, we applied an adversarial model, by varying the parameters of this model, we can generate a number of different adversarial strategies (e.g., always return the wrong answer, return a random answer, return a certain fraction of the correct answer, some of the adversaries return exactly the same answers, some adversaries return the perfectly colluding answers, assign each adversary with every task, assign each adversary with every task with a fixed probability). The details about the adversarial model as well as a definition of these parameters appears in Supplementary Sec. C.

Each graph in Figure 1 shows what happens when we vary one parameter. It can be seen that the M-MSR algorithm strongly outperforms the baseline methods on various datasets and under almost all of the adversarial strategies. Besides, the most damaging strategy in terms of reducing prediction error across the methods we tried seems to return a correct answer a fraction $q < 1/2$ of the time and an incorrect answer $1 - q$ of the time, and the adversaries should locate at the central places and be highly dependent with each other.

**Experiments on real data.** We implemented similar experiments on 17 publicly available data sets that are commonly used to evaluate the crowdsourcing algorithms. A detailed discussion of all the datasets can be found in Supplementary Sec. E, and the details of the how the experiments were conducted can be found in supplementary Sec. D. As shown in Figure 2 and Figure 6 (Supplementary Sec. D), the M-MSR algorithm consistently outperforms all the baseline methods. In particular, when the number of the corrupted workers increases, the prediction error of M-MSR algorithm maintains the smallest on almost every dataset.

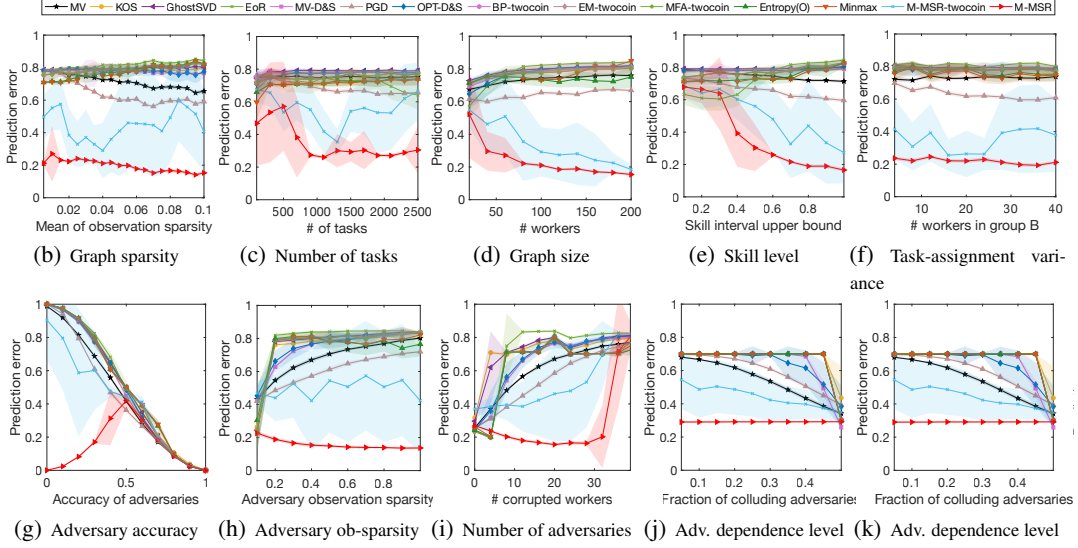

(b) Graph sparsity    (c) Number of tasks    (d) Graph size    (e) Skill level    (f) Task-assignment variance

(g) Adversary accuracy    (h) Adversary ob-sparsity    (i) Number of adversaries    (j) Adv. dependence level    (k) Adv. dependence level

Figure 1: Experiments on synthetic data (see Supplementary Section C for a full explanation).

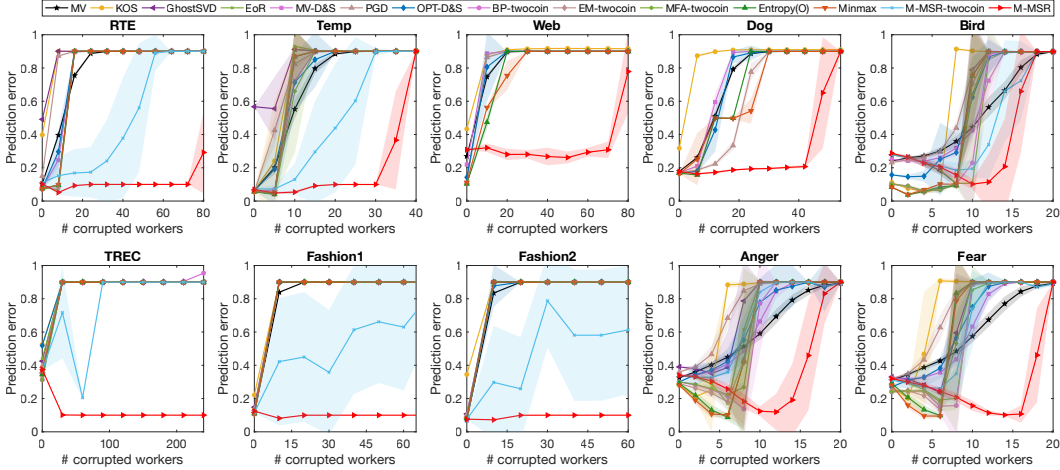

Figure 2: Experimental results on real datasets (see Supplementary Section D for a full explanation).

The M-MSR algorithm performs especially well on large datasets like RTE, Temp, TREC, Fashion1 and Fashion2. It is the only method which can handle around $\frac{n}{2}$ ($n$ is the number of the total workers) corrupted workers on these datasets. Out of 17 real datasets, our algorithm is the best on 16 of them. The only exception is dataset Surprise ( Figure 6 in Supplementary Sec. D) – the reason is that the "normal" workers on this dataset do not appear to be reliable. Besides, on some of the original real datasets, i.e., no adversaries are introduced, M-MSR algorithm is not the best one among all the baseline methods. This shows that the superiority of the M-MSR algorithm is mainly on large datasets in adversarial setting.

## 6.1 Exact Recovery

We compare the M-MSR algorithm with PCA and RPCA algorithms in [9]. We study the recovery rate of the rank-one matrices with different level of noises. The experiment results are shown in Figure 3 (the details of this experiment can be found in Supplementary Sec. F). It can be seen that our algorithm strongly outperforms these methods for robust rank-one matrix completion. We also note that the M-MSR algorithm is very efficient: when the dimension of the matrices increases from 10 to 1000, the running time of M-MSR increases from 0.01 seconds to 0.42 seconds while RPCA increases

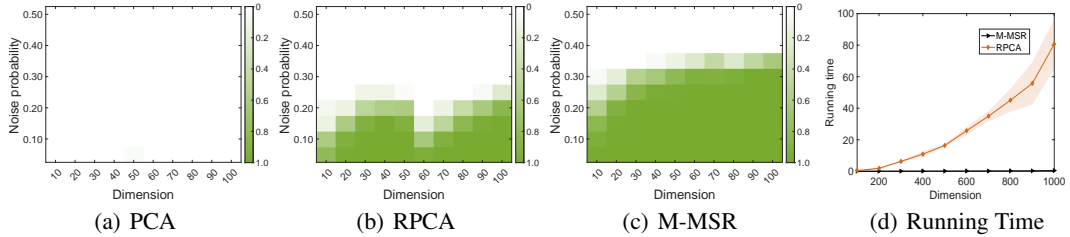

|(a) PCA|(b) RPCA|(c) M-MSR|(d) Running Time|

Figure 3: Experimental results of exact recovery experiment. (a) Recovery rate heatmap of subgradient method for PCA (The intensity of the color is proportional to the recovery rate). (b) Recovery rate heatmap of subgradient method for RPCA (c) The recovery rate heatmap for M-MSR (d) Running time of the subgradient method for RPCA and running time for M-MSR (for each dimension, the average running time and the standard deviation confidence interval over 100 independent trials are shown). Details are provided in Supplementary Section D.

from 0.46 seconds to 80.62 seconds. This is an additional advantage of the M-MSR algorithm when dealing with large datasets.

## 7 Discussion and Conclusions

We studied a crowdsourcing model with (i) The presence of users who might choose adversarial responses (ii) General worker-task assignment sets resulting in arbitrary interaction graphs $G(\Omega)$ among workers. Because approaches based on sparse recovery are not able to handle arbitrary $A$, we proposed a new algorithm, M-MSR, for skill determination (and consequently prediction) in this context. Our algorithm is based on a connection to the robust rank-1 matrix completion.

Our main results are: (i) A necessary and sufficient condition for our algorithm to work on any graph, and a proof that our algorithm is optimal on random graphs (ii) An empirical evaluation which shows that our algorithm outperforms existing methods on both synthetic and real data sets.

Future work will analyze M-MSR when the graph is partially random. While some scenarios, like Amazon's Mechanical Turk, allow any set $A$ to be specified, other practical scenarios do not; consider, for example, estimation of item quality from online ratings (e.g., Yelp), where the assignment of users to items is not random. Adversarial interactions are particularly important in this context, as business owners might be tempted to skew the ratings by leaving reviews from fake accounts.

Theorem 2 provides a bound to how many adversaries can be tolerated in this setting, and this bound will be quite good as long as the underlying graph is dense. For a sparse graph, however, this is not the case: in that case, Theorem 2 could fail to guarantee that even a small number of adversaries can't skew the result. One possibility is to strategically add more random edges (by giving users suggestions of items to rate) to make the resulting graph $2F + 1$-robust for a large $F$. Subsequent work could consider how well such schemes perform both in theory and in practice.

## Broader Impact

In this work, we provide a new robust matrix completion methods which can make recommendation systems more accurate in the presence of spam. This can benefit users of platforms like Amazon Mechanical Turk and Yelp.

In terms of negative impact, our work could allow the same platforms to learn more about the preferences of their users. It is possible that this data could be leaked, resulting in privacy loss.

## Acknowledgments and Disclosure of Funding

This work is supported by NSF awards 1914792 and 1933027.

## Footnotes

[1]The code is available on https://github.com/maqqbu/MMSR

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
