[Supplementary Material]

# A   Baselines

In this section, we describe all the methods used as baselines for comparisons.

## A.1   Crowdsoucing

- **Majority Voting (MV)** is a simple method where the true label of the tasks are estimated via the majority voting among the workers.
- **KOS** algorithm [18] is an approach for multi-class crowdsourcing problem with D&S model. The algorithm based on the assumption that the tasks are assigned to workers according to a random regular bipartite graph. To estimate the true labels, the $k$-class labeing problem is converted to a $k - 1$ binary labeling problems. Then these binary problems are iteratively solved via obtaining low-rank approximation of appropriate matrices. Though this algorithm requires specific constraints for the task assignment matrix, it can achieve good redundancy-accuracy trade-off.
- **Ghost-SVD** [11] algorithm considers the binary labeling problem based on single-coin D&S model. The true labels and the error probabilities of the workers are estimated via conducting the Singular Value Decomposition (SVD) to the observation matrix. This algorithm assumes that the probability error of one specific worker is smaller than 0.5. This algorithm has been proved to learn the true labels of the tasks with bounded error. However, this bound only works for the cases that the observation matrices are dense matrices.
- **Eigenvectors of Ratio (EoR)** [5] algorithm also considers the binary labeling problem which satisfies single-coin D&S model. Unlike Ghost-SVD and KOS approaches, this algorithm allows the task-assignment matrices to be arbitrary. The algorithm also applies a SVD-based method to obtain the estimation for the true labels and the workers' reliability. This algorithm has been proved to have an improved error-bound guarantee for single-coin crowdsourcing model than the previous methods.
- **EM algorithm (MV-D&S and OPT-D&S)** [41] is a two stage algorithm for multi-class crowdsourcing problem based on D&S model. In the first stage, the probability parameters of the D&S model are estimated by some approaches. In the second stage, the estimation of the parameters are refined by the standard EM algorithm (the results of stage 1 are used as an initialization). For MV-D&S, the majority voting method is used to get the initial parameter estimation in the first stage. For OPT-D&S, a spectral method is employed to obtain the initial parameter estimation in the first stage.
- **Projected Gradient Descent (PGD)** [30]. In [30], the skill estimation of the single-coin D&S model is formulated as a rank-one correlation-matrix completion problem. PGD approach updates the skill level of the workers via solving the following optimization problem with the conventional projected gradient descent algorithm with fixed stepsize.

$$\arg \min_{x \in [-1,+1]^W} \frac{1}{2} \sum_{(i,j) \in \Omega} N_{ij}(\tilde{C}_{ij} - x_i x_j)^2,$$

  where $W$, $N_{ij}$ and $\tilde{C}_{ij}$ are defined as in section 3, $x$ represents the skill level vector.
- **Variational Approaches (BP-twocoin, MFA-twocoin, EM-twocoin)** [28] address the crowdsourcing problems by using the tools and concepts from variational inference methods for graphical models. BP algorithm is a belief-propagation-based method, via choosing specific prior distributions of the workers' abilities, this algorithm can be reduced to KOS or majority voting. The BP algorithm can also be extended to more complicated models. MFA algorithm is a mean field algorithm which closely related to EM algorithm. In this work, we just consider the two coin version of these algorithms.
- **Regularized Minimax Conditional Entropy (Entropy(O))** [44] algorithm considers the crowdourcing classification problems with noises. In this algorithm, the confusion matrices of the workers are estimated as a minimax conditional entropy problem subject worker and items constraints that the workers can distinguish between classes which are far away from each other better than the ones which are adjacent.
- **Minimax Entropy Learning from Crowds (Minmax)** [42] algorithm also consider the crowdourcing classification problems with noises. The difference is that the Minmax

algorithm estimates the confusion matrices via maximizing the entropy of the probability distribution over workers. Besides, they give prediction for the tasks by minimizing the KL divergence between the probability distribution and the unknown truth.

## A.2 Exact Recovery

- **Robust Principle Component Analysis (RPCA)** [9] considers non-negative rank-one matrix completion problem. The rank-one matrix can be reconstructed via solving the following optimization problem.

$$\min_{\boldsymbol{u}\in\mathbb{R}^m_+, \boldsymbol{v}\in\mathbb{R}^n_+} \|\mathcal{P}_\Omega(X - \boldsymbol{u}\boldsymbol{v}^\top)\|_1 + R_\beta(\boldsymbol{u}, \boldsymbol{v}), \tag{12}$$

  where $X$ is the matrix to be recovered, $\Omega$ represents the set of the locations of the observed entries, $R_\beta(\boldsymbol{u}, \boldsymbol{v}) = \alpha|\boldsymbol{u}^\top\boldsymbol{u} - \boldsymbol{v}\boldsymbol{v}^\top|$ is a regularization term. It has been proved that [9] does not have local minimum when some specific conditions satisfied. RPCA algorithm can also recover the matrix $X$ when there exists some sparse noises. In [9], the optimization problem (12) is solved via the subgradient descent method, whereas other optimization algorithms can also be used to solve it. In our experiment, we used subgradient descent method with diminishing step size to get the optimal solution of (12).

- **Principle Component Analysis (PCA)** [9] considers the same problem as RPCA. PCA algorithm recover the rank-one matrix $X$ via solving the following smooth optimization problem,

$$\min_{\boldsymbol{u}\in\mathbb{R}^m_+, \boldsymbol{v}\in\mathbb{R}^n_+} \|\mathcal{P}_\Omega(X - \boldsymbol{u}\boldsymbol{v}^\top)\|_F^2. \tag{13}$$

  In our experiment, we employed gradient descent with fixed stepsize to optimize (13).

# B  Additional baselines and the two-coin model

The M-MSR method can be extended to solve rank-2 matrix completion problems with corruptions. Suppose the rank-2 matrix we aim to recover is $X \in \mathbb{R}^{n\times m}$, $\Omega$ is the set of the revealed locations. Let $X = \boldsymbol{u}\boldsymbol{v}^T$, where $\boldsymbol{u} \in \mathbb{R}^{n\times 2}$, $\boldsymbol{v} \in \mathbb{R}^{m\times 2}$, we want to find $\boldsymbol{u}$ and $\boldsymbol{v}$. To deal with the corrupted entries, we define

$$J_i(t) = \left\{ \frac{X_{ij}}{\|v_j(t)\|} \middle| (i, j) \in \Omega \right\}.$$

We will then set $R_i(t)$ to be the set of nodes with the $F$ largest and smallest values in $J_i(t)$ (if there are fewer than $F$ values strictly smaller/larger than $u_i(t)$, then $R_i(t)$ contains the ones that are strictly smaller/larger than $u_i(t)$); the quantities $J_j'(t), R_j'(t)$ are defined similarly for nodes $j \in V_v$. The extended algorithm is presented next.

Now cosider the two-coin model of the crowdsourcing problem, where the ability of worker $i$ is specified by two parameters $s_i$, $t_i$:

$$s_j = \text{Prob}[Y_{ij} = +1|g_j = +1], \qquad t_j = \text{Prob}[Y_{ij} = -1|g_j = -1]. \tag{16}$$

Suppose assignment of tasks to workers is random and the proportion of tasks which have an answer of $+1$ is $p$. Suppose $p$ is known. Let $K_{ab}$ be the set of tasks assigned to both workers $a$ and $b$. Then

$$\frac{1}{K_{ab}} \sum_{j\in K_{ab}} Y_{aj}Y_{bj} \approx E[Y_{aj}Y_{bj}]$$

$$= p\left(s_a s_b + (1-s_a)(1-s_b)\right) + (1-p)(t_a t_b + (1-t_a)(1-t_b))$$

$$= p\left(\frac{1}{2} + \frac{1}{2}(2s_a - 1)(2s_b - 1)\right) + (1-p)\left(\frac{1}{2} + \frac{1}{2}(2t_a - 1)(2t_b - 1)\right)$$

$$= \frac{1}{2} + \frac{p}{2}(2s_a - 1)(2s_b - 1) + \frac{1-p}{2}(2t_a - 1)(2t_b - 1).$$

**Algorithm 2** M-MSR-twocoin

**Input:** Positive matrix $X$, set $\Omega$, $F$ and $\boldsymbol{v}(0) > 0$
**Output:** $\hat{X} = \boldsymbol{u}(T)\boldsymbol{v}(T)^\top$
1: **for** $t = 1, 2, \ldots, T$ **do**
2:     For each $i = 1, \cdots, m$, let

$$u_i(t+1) = \arg\min_u \sum_{j \in \Omega_i \setminus R_i(t)} (u^\top v_j(t) - X_{ij})^2; \tag{14}$$

    where $u_i \in \mathbb{R}^2$ denotes the $i$th row of $\boldsymbol{u}$.
3:     For each $j = 1, \cdots, n$, let

$$v_j(t+1) = \arg\min_v \sum_{i \in \Omega'_j \setminus R'_j(t)} (u_i(t)^\top v - X_{ij})^2, \tag{15}$$

    and $v_j \in \mathbb{R}^2$ denotes the $j$th row of $\boldsymbol{v}$.
4: **end for**
5: **return** $\boldsymbol{u}(t) = [u_1(t), u_2(t), \cdots, u_m(t)]^\top$, $\boldsymbol{v}(t) = [v_1(t), v_2(t), \cdots, v_n(t)]^\top$

In particular, we have

$$\left[ \frac{1}{K_{ab}} \sum_{j \in K_{ab}} Y_{aj}Y_{bj} - \frac{1}{2} \right]_{ab} \approx \text{ rank-2 matrix}$$

In this case, we can apply the M-MSR-twocoin algorithm to estimate the skill level of the workers in two-coin model, and further give predictions for the tasks.

## C Synthetic Experiments

The purpose of this section is to discuss the details of the synthetic experiments as shown in Figure 1. We will analyze the impact of graphs, task assignments, skill level and different adversarial strategies to the performance of the M-MSR algorithm as well as other baseline methods. Specifically, we will experiment with increasing level of graph sparsity, number of tasks, graph size, average skill and decreasing level of tasks assignment variance.

For the adversarial model, we randomly choose a certain number of workers, and let these workers be adversaries. Then these adversaries will be evenly divided into some groups, and the members of the same group will produce exactly the same response for each task (for the tasks they are assigned with). In this case, the adversaries in the same group are no longer independent of each other. The answer set of each group will be generated randomly according to a given accuracy (we randomly choose a fraction of tasks and give correct answer for them, for others we give wrong answers). Each adversary will be assigned with every task according to a fixed probability (obs-sparsity), and then they will produce answers for the assigned tasks from the answer set. Hence, by varying the total number of the adversaries, the number of groups, the level of accuracy and obs-sparsity, we can generate a number of different adversarial strategies.

For each experiment, we vary one parameter while keep the others fixed. The detailed information of the original dataset we apply is given in Table 1. Among them, the skill distribution represents the grid from which we choose the skill level of each normal worker uniformly at random. The group-B-#tasks is the number of tasks in group B, the details will be introduced in the "impact of task assignment variance". The obs-sparsity represents the probability that each task can be assigned to a specific worker.

Table 1: Synthetic dataset: characteristics values of the original dataset

| #tasks | #workers | #class | skill | group-B-#tasks | ave.obs-sparsity | #corruptions | adversary ACC | adversary obs-sparsity | #ad-groups |
|---|---|---|---|---|---|---|---|---|---|
| 1600 | 80 | 2 | $[-0.1, 0.7]$ | 20 | 0.04 | 20 | 0.3 | 0.4 | 5 |

In the synthetic experiments, we make one minor modification to the M-MSR algorithm, i.e. after the algorithm converging, we will project the obtained $s_i$ which away from cube $[-\frac{1}{M-1} + \frac{1}{\sqrt{N_i}}, 1 - \frac{1}{\sqrt{N_i}}]$ onto it, where $N_i$ is the number of the tasks assigned to worker $i$. The reason for this modification is to stay away from the boundary of the hypecube where the weight log-odds function is changing very rapidly. For the convenience of the analysis, we define

$$
\begin{aligned}
\hat{C}_{ij} &= \frac{M}{M-1}\widetilde{C}_{ij} - \frac{1}{M-1} \\
&= \frac{M}{M-1}\frac{1}{N_{ij}}\sum_{t|(i,t),(j,t)\in A}\langle Y_{i,t}, Y_{j,t}\rangle - \frac{1}{M-1},
\end{aligned} \tag{17}
$$

then the skill vector $s$ in the RHS of (3) can be estimated by reconstructing the matrix $\hat{C}$.

Next, we will provide the details of each experiment and discuss the experiment results of each graph in Figure 1 respectively.

**Impact of graph sparsity**: Figure 1(b) shows the experimental results of varying the sparsity of the interaction graph. The interaction graph sparsity implies the sparsity of $\mathcal{G}(\Omega)$ corresponding to the rank-one matrix $\hat{C}$ in (17), which has a close connection to the connectedness and robustness of the graph. To see the impact of the interaction graph sparsity to the performance of different methods, we increase the observation sparsity level (the probability such that one element of $Y_{w,t}$ is nonzero) which in turn increases the sparsity of the workers' interaction graph. The relationship of the graph sparsity and observation sparsity is shown in Figure 4(a), where we generate $Y_{w,t}$ randomly according to the observation sparsity and then observe the graph sparsity corresponds to $\hat{C}$. It can be seen when the observation sparsity increases from 0 to 0.08, the graph sparsity increases from 0 to around 1, and after that, the graph sparsity maintains approximately 1. Therefore, we can vary the mean of the observation sparsity from 0 to 0.1 to see the impact of the graph sparsity to the experimental results.

It can be observed when the mean of the observation sparsity varies from 0 to 0.1, the prediction errors of the baseline methods except the M-MSR algorithm keeps greater than 0.6. The is because the adversaries are dominating the prediction results of these baseline methods. Why these adversaries can dominate the prediction results? There are three reasons. First, $1/4$ of the workers are adversaries with accuracy 0.3 in this experiment, and each of them was assigned with around $40\%$ tasks (the default value of the adversary obs-sparsity is 0.4). In other words, among all of the collected answers, a large fraction of them come from the adversaries and are incorrect. Next, the obs-sparsity of the adversaries is 0.4 implies that they have common tasks with almost all of the normal workers, hence the adversaries are located in the central places of the graph, and their behaviors can have a great impact to the prediction results. Moreover, these adversaries are not following the single-coin D&S model, they are highly dependent with other, therefore the baseline methods can not correctly estimated the behaviors of the adversaries.

However, it can also be observed that the prediction error of the M-MSR algorithm maintains smaller than 0.2 in most time. This is because that the true skill level of the normal workers can be correctly estimated by the M-MSR algorithm. More importantly, though the adversaries are not following the single-coin D&S model, they will be assigned with the negative skill levels when applying the M-MSR algorithm. Consider an adversary $i$, it is very likely that the value of $\widetilde{C}_{ij}$ will be quite small if $j$ is a normal worker. Meanwhile, as we have discussed, adversary $i$ is located in the central place, which means it can be connected to almost every normal workers. Thus, in the $i$th row and column of $\widetilde{C}$, the majority of elements will be very small, and according to (17), the corresponding elements in $\hat{C}$ will be negative. Though the elements in $\hat{C}$ related to the correlation with other adversaries can be positive or even be 1, the M-MSR algorithm can still assign adversary $i$ with a negative skill level $s_i$. This negative $s_i$ can produce a negative weight for the wrong answers from $i$ in the prediction. Therefore, the M-MSR algorithm can give accurate predictions in such adversarial scenario.

Besides, the prediction error of the M-MSR algorithm tends to decrease when the graph sparsity increases. This is because the robustness of $\mathcal{G}(\Omega)$ increases when the graph are becoming more dense. In this case, according to Theorem 2, it is more likely that the true skill level of the normal workers can be correctly estimated, and the adversaries will be assigned with smaller negative weights, hence the prediction accuracy can be improved.

**Impact of the number of tasks**: Figure 1(c) shows the experimental results of varying the the number of the tasks. In this experiment, we vary the number of tasks from 100 to 2500. It can be observed that the prediction accuracy of M-MSR algorithm increases when the number of the tasks increases. The reason for this phenomenon is that the noise level of the empirical estimation of $\hat{C}$ is decreasing. When the number of the tasks is small, even for the normal workers, the corresponding elements in $\hat{C}$ can be regarded as corrupted. The M-MSR algorithm can not work very well when there exists no reliable workers. Besides, the prediction error of the baseline methods keeps greater than 0.6 in most of the time, the reason is similar as in the experiment of the graph sparsity.

**Impact of the graph size**: Figure 1(d) shows the experimental results of varying the level of the graph size. Since the graph size is associated with the number of the workers, we can vary the number of the workers to see the impact of the graph size to the performance. From Figure 1(d) , we can see when the number of workers is small, the standard deviation of the prediction error for M-MSR algorithm is large. This is because we have assumed the workers' skill level distribution is $[-0.1, 0.7]$, and we randomly corrupt $\frac{1}{4}$ of the normal workers each time. When the number of the workers is small, the skill levels of these normal workers may be quite different. If different normal workers are corrupted, the skill level of the remaining normal workers can be different, and the corresponding prediction error can also be different. When the number of the workers is large, it is more likely that the remaining normal workers can cover all possible skill level, hence the prediction error tends to be steady. For the other baseline methods, due to the reason we analyzed, the performance is dominated by adversaries, hence the difference of the number of the works does not have much impact to the prediction error.

**Impact of different skill-distribution**: Figure 1(e) shows the experimental results when varying the skill level of the normal workers. In the synthetic experiments, we assign the workers skill level uniformly at random on a grid. To see the impact of the skill level to the prediction accuracy, we fix the lower bound of the grid as -0.1 and vary the upper bound of the grid from 0.1 to 1. It can be observed in Figure 1(e) that the prediction error of the M-MSR algorithm decreases when the upper bound of the skill level increases. When the average skill level of the normal workers is low, the corresponding elements in $\widetilde{C}$ can be relatively large as both the adversaries and normal workers are providing the wrong answers. As a consequence, the estimated skill level for the adversaries may be larger, or even positive. Hence the prediction error can be large. Meanwhile, the performance of the other algorithms keeps greater than 0.6. The reason is that these algorithms are still dominated by the adversaries, hence the performance does not change when the skill level of the normal workers changes.

**Impact of the tasks assignment variance**: Figure 4(b) shows the experimental results when varying the level of the tasks assignment variance. For a real-world crowdsourcing dataset, it is highly possible that the number of the tasks assigned to different workers can be different. Hence, we want to see the impact of the tasks assignment variance to the prediction accuracy. However, it is not easy to directly vary the variance while keep the total number of the answers fixed. To solve this issue, we vary another parameter which is closely related to the task assignment variance. We divide the workers into two groups, i.e., group A and group B. The workers in group A and group B provide approximately the same number of answers in total (we fix the observation sparsity sum in the two groups), and each worker in the same group will be assigned with approximately the same number of tasks (the observation sparsity for each worker is the average of the fixed observation sparsity sum). As a consequence, when we increase the number of workers in group B, the tasks assignment variance will decrease. This can be observed in Figure 4(b) which shows the results where we vary the number of the workers in group B and observe the task assignment standard deviation.

From Figure 1(f) , we can see the task assignment variance does not have much impact to the prediction accuracy for all the algorithms. When varying the level of the task assignment variance, the prediction error of the M-MSR algorithm maintains around 0.2 while the prediction error of other baseline methods maintains greater than 0.6.

**Impact of adversary accuracy**: Figure 1(g) shows the experimental results of varying the level of the adversary accuracy. It can be observed that the prediction error of the baseline methods except the M-MSR decreases as the accuracy of the adversaries increases. The prediction error of these algorithms are approximately equal to the error of the adversaries. This phenomenon largely results from the fact that the adversaries are dominating the prediction results.

(a) The relationship of graph sparsity and observation sparsity. The details can be found in "Impact of Graph Sparsity".

(b) The relationship of task assignment variance and number of workers in group B. The details can be found in "Impact of tasks assignment variance".

Figure 4: The parameter design analysis

However, the situation is quite different for the M-MSR algorithm. When the accuracy of the adversaries increases from 0 to 0.5, the prediction error of the M-MSR algorithm increases from 0 to around 0.35, and then the predction error of the M-MSR algorithm decreases again to 0 when the accuracy increases from 0.5 to 1. A major factor which contributes to such phenomenon is that the adversaries were assigned with a negative skill level $s_i$ when the adversary accuracy is smaller than 0.5, and they were assigned with positive skill level $s_i$ when the accuracy is greater than 0.5. When the adversary accuracy is smaller than 0.5, the adversaries can produce more wrong answers than correct answers. As a consequence, in the correlation matrix $\widetilde{C}$, the elements which corresponds to the correlation between the adversaries and the normal workers have relatively small values. As we analyzed in the experiment of the graph sparsity, the adversaries can be assigned with negative skill level by M-MSR algorithm. In this case, the smaller the adversary accuracy is, the smaller the skill level will be. When the accuracy of the adversaries is greater than 0.5, both the adversaries and the normal workers will produce more correct answers. Thus, it is very likely that the elements in $\widetilde{C}$ corresponds to the correlation between the adversaries and the normal workers have the values greater than 0.5, which can further lead to the estimated skill level of the adversaries be positive. In such scenario, the more accurate the adversaries are, the more accurate the prediction results given by M-MSR will be.

**Impact of adversary observation sparsity**: Figure 1(h) shows the experimental results when varying the observation sparsity of the adversaries. It can be observed that the prediction error of the baseline methods except the M-MSR increases when the adversary obs-sparsity increases. This largely results from the fact the number of answers from the adversaries (the majority of them are incorrect) are increasing when the obs-sparsity increases. Another factor contributes to this phenomenon is that the adversaries can be connected to more normal workers as the obs-sparsity increases, which can make the adversaries have greater impact to the prediction results. Particularly, when the obs-sparsity of the adversaries is approximately equal to the normal workers, i.e., it is 0.05, the prediction error of almost all of the crowdsourcing methods are less than 0.5. In other words, when the adversaries are assigned with very small number of tasks, and they are not located in the central positions, then these adversaries can not greatly damage the performance of these methods. However, once the obs-sparsity of the adversaries increases to 0.1, the adversaries again can dominate the prediction results of these methods.

Next consider the M-MSR algorithm. The prediction error of the M-MSR algorithm maintains smaller than 0.2 as we analyzed in the graph sparsity experiments. Meanwhile, it can be seen that the prediction error of the M-MSR algorithm tends to decrease when the adversary obs-sparsity increases. One possible cause is that the adversaries can be connected to more normal neighbors which allow the M-MSR algorithm to give smaller negative skill estimation for the adversaries. Another reason maybe the weight of the wrong answer tends to be smaller when more adversaries are involved in the sum of (11) as the adversaries are assigned with negative weights.

**Impact of the number of the adversaries** Figure 1(i) shows the experimental results when varying the number of the corruptions. In this experiment, we vary the number of the workers from 0 to 40 (the total number of the workers is 80). It can be observed the prediction errors of all the algorithms

increase when the number of the corruptions increases. The M-MSR algorithm is the only method which can handle the corruptions up to 28. The reasons are similar to our previous analyses.

**Impact of adversary dependence level** To study the impact of the adversary dependence level, we implemented two different experiments. Figure 1(j) shows the experimental results of the first experiment, where we vary the number of adversary groups. In our adversarial model, the members in the same group produce exactly the same response to every task. Thus, the dependence level between the adversaries decreases as the number of the adversary groups increases. When the number of the groups is 20, each of the adversary is independent of each other (the default number of the corrupted workers is 20). It can be observed that the prediction errors of the MultiSPA, MultiSPA-EM, and MultiSPA-KL decreases as the number of the adversary groups increases. When the number of the groups is 20, the prediction error of the MultiSPA-EM decrease to be smaller than 0.2. Such phenomenon is understandable, if every adversary is independent of each other, then the adversaries satisfies the requirement of the single-coin model, and they will just be normal workers with low skill level. It is possible that some methods can handle such kind of adversaries. Meanwhile, it can be observed that the prediction error of the M-MSR algorithm maintains to be smaller than 0.2 in almost all of the time.

Figure 1(k) shows the experimental results of the second experiment, where the 20 adversaries are divided into two groups. The members in the first group work as before, they produce the same response for every task, the accuracy of the answer set is also 0.7, and the obs-sparsity of the adversaries is 0.4. However, the answer set of the second adversary group is set to be perfectly colluding with the answer set of the first group, i.e., the accuracy of the second group is 0.7. Then similarly, the adversaries in the second group are also assigned with tasks with probability 0.4, and each of them will give answers for these tasks from the answer set. In this experiment, we vary the number of the adversaries in second group from 0 to 10, which is equivalent to vary the fraction of the second group over the total number of the adversaries from 0 to 0.5. It can be observed when this fraction increases, the prediction error of the baseline methods except the M-MSR algorithm decrease. This phenomenon largely results from the fact that the adversaries in the second group can produce more accurate answers than the ones in the first group. Besides, it can be seen that the prediction error of the M-MSR algorithm keeps to be 0.3 when the fraction of the second group changes. This is because the M-MSR algorithm give the adversaries in the first group the negative skill level estimations and give the ones in the second group the positive skill level estimations. In fact, all of these skill estimations will lead to the prediction results be dominated by the answers of the second group, where the error is 0.3.

From the two experiments, we can see returning the same answers can reduce the prediction error across the crowdsourcing methods more than returning colluding answers. Besides, the higher the dependence level of the adversaries is, the more damaging the adversarial strategy will be.

|   | A | B | C | D | E | F |
|---|---|---|---|---|---|---|
| **A** |   | 1 | 0.08 | 0.08 | 0.08 | 0.08 |
| **B** | 1 |   | 0.08 | 0.08 | 0.08 | 0.08 |
| **C** | 0.08 | 0.08 |   | 0.64 |   |   |
| **D** | 0.08 | 0.08 | 0.64 |   | 0.64 | 0.64 |
| **E** | 0.08 | 0.08 |   | 0.64 |   | 0.64 |
| **F** | 0.08 | 0.08 |   | 0.64 | 0.64 |   |

Figure 5: An illustration example of the covariance matrix $\widetilde{C}$ when there exists adversaries as in our adversarial model. The elements colored with red are corrupted, and the green ones are normal elements, the white ones are missing elements. A and B represent adversaries which produce the same response for every task, and the accuracy of their answer set is 0.1. C, D, E, and F represent normal workers with skill level 0.8. The number of tasks are large enough. For such a covariance matrix $\widetilde{C}$, the M-MSR algorithm can correctly estimate the true skill level of the normal workers and give negative skill estimation for the adversaries, and further provide accurate prediction results. However, for algorithms like PGD, the prediction is not accurate due to the impact of the corruptions.

# D   Real dataset Experiments

In this section, we provide the implementation details and results analyses about the real dataset crowdsourcing experiments as shown in Figure 2 and Figure 6.

In the real dataset experiments, all of the datasets come with ground truth label for each task (we remove the tasks with missing ground truth labels). In order to improve the efficiency of the experiments and reduce the sparseness, we remove the workers who provided less than 10 answers for each dataset. Since the labels of the datasets Emotion (Joy, Surprise, Anger, Disgust, Sadness, Fear) and Valence are numerical values, we transform these numerical labels to binary labels according to different partitions to the label range. The characteristic values and detailed introduction of these datasets are provided in Supplementary Sec E. Moreover, since Ghost-SVD algorithm and EoR algorithm work only for binary tasks, they will not be evaluated on the multiple-class datasets Adult2, Dog, Web and WSD. Besides, we will also apply the same projection strategy as in synthetic experiments for the M-MSR algorithm, i.e. after the algorithm converging, we will project the obtained $s_i$ which away from cube $[-\frac{1}{M-1} + \frac{1}{\sqrt{N_i}}, 1 - \frac{1}{\sqrt{N_i}}]$ onto it.

In the real data experiments, the adversaries are also randomly corrupted, and they follow the same adversarial model as in synthetic experiments. Here, we set the number of the adversary groups is 1, the accuracy of the adversaries is 0.1, and the observation sparsity is 0.5. According to the results of the synthetic experiments, the most damaging strategy is to return a correct answer a fraction $q < 1/2$ of the time and an incorrect answer $1 - q$ of the time, and the adversaries should locate at the central places and be highly dependent with each other. Follow this idea, we set the parameters of the adversarial model as listed above. Moreover, for each dataset, we vary the number of the adversaries from 0 to around half of the number of the workers and then observe the corresponding prediction error of the crowdsourcing methods. The results of the real dataset experiments are given in Figure 2 and Figure 6.

It can be seen that the prediction error of almost all the algorithms on each dataset converge to 0.9 when the number of the corruptions increases. The reason is that the prediction error of the adversaries is 0.9 and they will gradually dominate the prediction results when their number grows. Besides, on large datasets Fashion1, Fashion2, TREC, Temp and RTE, the prediction error of the baseline methods except the M-MSR increases to 0.9 rapidly when the number of the adversaries increases. This largely results from the fact that the number of the tasks of these datasets is large. When the number of the adversaries increases, the fraction of the answers from the adversaries will increase rapidly. In other words, the fraction of the wrong answers among all of the answers increases rapidly, which can lead to such phenomenon. However, on these datasets, the prediction error of the M-MSR algorithm maintains to be 0.1 in most of the time. This is because the noise level of the covariance matrix $\widetilde{C}$ is low on these datasets. In this case, as we analyzed in "Impact of graph sparsity", it is more likely that the true skill level of the normal workers can be correctly estimated and the adversaries can be assigned with negative skill estimations. According to prediction rule (11), such skill estimation can lead to the prediction results be opposite to the answer set of the adversaries. Therefore, the prediction error of the M-MSR algorithm can be 0.1 in most of the time. When the number of the adversaries increases to around a half of the total numbers, the M-MSR algorithm can not give negative skill estimation for the adversaries and hence its prediction error will also converge to 0.9.

Moreover, out of 17 real datasets, our algorithm is the best on 16 of them. The only exception is dataset Surprise (Figure 6) – the reason is that the "normal" workers on this dataset do not appear to be reliable. We can see from Table 2 that the average error probability of the normal workers is greater than $0.5$ for this dataset, which means the average skill level of the normal workers is negative. From Figure 1(e) and the analysis in synthetic experiments, we can see such phenomenon is normal. Though our algorithm can eliminate the impact of the adversaries, we can not give accurate predictions if the remaining normal workers are not reliable, either. Besides, the Surprise dataset is a relatively small dataset, which implies that the noise level of the $\widetilde{C}$ can be large. This can further reduce the prediction accuracy of the M-MSR algorithm.

Figure 6: Experimental results of real data as well as a two-coin synthetic dataset. The synthetic dataset is created following two-coin model rule, where a worker $j$ is parametrized $s_j$ and $t_j$, which are defined as in (16). In this experiment, we let prior probability $p = 0.5$, and choose $s_j$ and $t_j$ uniformly in $[0.5, 1]$.

# E  Datasets

In this section, we introduce the real datasets we applied in the crowdsourcing experiments. We employ 17 public real datasets to evaluate the effectiveness of the M-MSR algorithm and the baseline methods in crowdsourcing experiments. The followings are the brief introduction of these datasets.

- **Fashion (Fashion1, Fashion2)** [2] [29] is a fashion-focused Creative Commons images dataset associated with two different labels. Fashion1 dataset corresponds to the first label, which indicates if an image is fashion-related or not. Fashion2 dataset corresponds to the second label, which indicates whether the fashion category of the image can correctly characterize the content in the image. The ground truth and the labels of the dataset was collected on Amazon Mechanical Turk (MTurk) platform. Fashion1 contains 13727 labels for 4711 images which are provided by 202 workers. Fashion2 contains 13474 labels for 4710 images which are provided by 208 workers.

- **TREC**[3] [24] is a binary-class dataset where the task is to judge the relevance of the documents. The dataset is provided in TREC 2011 crowddourcing track. There are 88385 labels collected from 762 workers for 19033 documents in total.

- **Waterbird Dataset (Bird)**[3] [38] is a binary-class dataset where the task is to identify whether an image contains a duck or not. There are 108 images in toal, and 4212 labels are collected from 39 workers. The labels are collected on MTurk platform.

- **Dog**[3] [7] is a multiclass-dataset where the task is to recognize a breed (out of Norfolk Terrier, Norwich Terrier, Irish Wolfhound, and Scottish Deerhound) for a given dog. There are 7354 labels collected from 52 workers for 807 documents in total.

- **Temporal Ordering (Temp)**[4] [36] is a binary-class dataset about the temporal ordering of event pairs. The workers are presented with event pairs and are asked to decide if the event described by the first verb occurs before the second one. The verb event pairs are extracted from [33] by Snow et. al.[36]. There are 4620 labels provided by 76 workers for 462 event pairs in total. The labels are collected on MTurk platform.

- **Recognizing Textual Entailment (RTE)**[4] [36] is a dataset where the workers are presented with two sentences in each example and are asked to decide whether the scond sentence can be inferred from the first one or not. These sentence pairs come from PASCAL Recognizing

[2]Available at `http://skulddata.cs.umass.edu/traces/mmsys/2013/fashion/`
[3]Available at `https://github.com/zhangyuc/SpectralMethodsMeetEM/tree/master/src`

Textual Entailment task [4]. There are 800 sentence pairs in total which are labeled by 80 workers on MTurk platform, and 8000 labels are collected.

- **Web Search Relevance Judging (Web)**[3] [43] is a multi-class datset where the task is to judge the relevance of query-URL pairs with a 5-level rating scale (from 1 to 5). There are 2665 query-URL pairs labeled by 177 workers, and the total number of the collected labels is 15567. The labels are collected on MTurk platform.

- **Word Sense Disambiguation (WSD)**[4] [36] is a dataset to identify the most appropriate sense (out of three given senses) of the word "president" in a given paragraph. These paragraph examples are sampled from SemEval Word Sense Disambiguation Lexical Sample task [32] by Snow et al. [36]. There 1770 labels collected for 177 examples from 10 workers on MTurk platform.

- **Emotions (Fear, Surprise, Sadness, Disgust, Joy, Anger, Valence)**[4] [36] is a group of datasets about ratings of different emotions for a given headline. There are six emotions datasets (fear, surprise, sadness, disgust, joy, anger) where the workers are asked to give numerical judgements in the interval $[0, 100]$ rating the headline for each emotion. Besides, there is a valence dataset where the workers give numerical rating in the interval $[-100, 100]$ which represents the overall positive or negative velence of the emotional content of the headline. The headlines are sampled from the SemEval-2007 Task 14 [37] by Snow et al. [36]. The labels are collected on the MTurk platform. There are 1000 labels for 100 headlines which are provided by 10 workers for each dataset. Since the labels of these datasets are numerical values, we convert them to binary-class datasets according to different partitions of the interval range. For emotions datsets, we let the rating value 0 represnt negative class and the rating interval $(0, 100]$ represent the negative class (0 means the corresponding emotion is not observed). For valence dataset, we segment the interval $[-100, 100]$ to $[-100, 0)$ (negative) and $[0, 100]$ (positive) respectively.

- **Adult2** [5] [16] is a multi-class dataset about the adult level of websites (G, PG, R and X). The labels are provided by workers on AMT platform. This dataset contains 3317 labels for 333 websites which are offered by 269 workers.

In our real dataset crowdsoucing experiments, we remove the workers who provide less than 10 labels for each dataset to reduce the sparsity of $\mathcal{G}(\Omega)$ as well as improve the efficiency. Table 2 shows the characteristic values of the real datasets after this change.

Table 2: Real data: characteristic values after removing workers who provide less than 10 labels

| Dataset | #workers | #tasks | #class | graph density | #crowdsourced labels(overall) | ave.(min/max) #labels/worker | ave.(min/max) #workers/tasks | average(min/max) prob. error |
|---------|----------|--------|--------|---------------|-------------------------------|------------------------------|------------------------------|------------------------------|
| Adult2 | 269 | 333 | 4 | 0.14 | 3317 | 12.3 (1/184) | 10.0 (1/21) | 0.35 (0.00/1.00) |
| Anger | 38 | 100 | 2 | 0.30 | 1000 | 26.3 (20/100) | 10 (10/10) | 0.35 (0.10/0.60) |
| Bird | 39 | 108 | 2 | 1.00 | 4212 | 108 (108/108) | 39 (39/39) | 0.36 (0.11/0.68) |
| Disgust | 38 | 100 | 2 | 0.30 | 1000 | 26.3 (20/100) | 10 (10/10) | 0.26 (0.05/0.50) |
| Dog | 109 | 807 | 4 | 0.58 | 8070 | 74.0 (1/345) | 10 (10/10) | 0.30 (0.00/1.00) |
| Fashion1 | 196 | 3742 | 2 | 0.07 | 10983 | 56.0 (1/962) | 2.9 (1/3) | 0.18 (0.00/1.00) |
| Fashion2 | 198 | 3601 | 2 | 0.07 | 10420 | 52.6 (1/925) | 2.9 (1/3) | 0.11 (0.00/1.00) |
| Fear | 38 | 100 | 2 | 0.30 | 1000 | 26.3 (20/100) | 10 (10/10) | 0.35 (0.10/0.80) |
| Joy | 38 | 100 | 2 | 0.30 | 1000 | 26.3(20/100) | 10 (10/10) | 0.43 (0.10/0.65) |
| RTE | 164 | 800 | 2 | 0.09 | 8000 | 48.8 (20/800) | 10 (10/10) | 0.16 (0.00/0.60) |
| Sadness | 38 | 100 | 2 | 0.30 | 1000 | 26.3 (20/100) | 10 (10/10) | 0.36 (0.15/0.65) |
| Surprise | 38 | 100 | 2 | 0.30 | 1000 | 26.3 (20/100) | 10 (10/10) | 0.51 (0.00/0.85) |
| TEMP | 76 | 462 | 2 | 0.25 | 4620 | 60.8 (10/462) | 10 (10/10) | 0.16 (0.00/0.60) |
| TREC | 677 | 2275 | 2 | 0.04 | 12863 | 19 (1/ 967) | 5.7 (1/10) | 0.32 (0.00/1.00) |
| Valence | 38 | 100 | 2 | 0.30 | 1000 | 26.3 (20/100) | 10 (10/10) | 0.34 (0.10/0.65) |
| Web | 176 | 2653 | 5 | 0.15 | 15539 | 88.3 (1/1225) | 5.9 (2/12) | 0.63 (0.00/1.00) |
| WSD | 34 | 177 | 3 | 0.44 | 1770 | 52.1 (17/177) | 10 (10/10) | 0.02 (0.00/0.17) |

## F  Further Experiments: Exact Recovery

The purpose of this section is to discuss the details of the exact recovery experiments as shown in Figure 3. For this experiment, we compare the proposed M-MSR algorithm to AN-RPCA, PCA algorithms in [9]. We consider thousands of randomly generated positive rank-1 matrix with different sizes and and different noise levels. The size of the matrices ranges from $10 \times 10$ to $100 \times 100$. The elements of $\mathbf{u}^*$ and $\mathbf{v}^*$ are uniformly chosen from the interval $[0, 2]$. Each element of the noise matrix $S$ is generated to be 200 with probability $p$ and 0 with probability $1 - p$. We assume that a matrix can be exactly recovered if $\|\mathbf{u}\mathbf{v}^\top - \mathbf{u}^*\mathbf{v}^{*\top}\|_F / \|\mathbf{u}^*\mathbf{v}^{*\top}\|_F \leq 10^{-4}$. For each dimension and noise probability, we generate 100 random matrices under such conditions and demonstrate its exact recovery rate. To improve the efficiency of the M-MSR algorithm, we did not adopt the random initialization in the exact recovery experiment. Instead, we choose arbitrary row of $X$, and complete the unobserved entries of this row with random positive constants, then let this row be $\mathbf{v}(0)$. In this case, part of the nodes in $\mathcal{G}(\Omega)$ have the same value corresponding to $k'_j = \frac{1}{a_i}$ in the beginning (as $v_j(0) = a_i b_j = \frac{b_j}{k'_j(0)}$), hence the consensus process can be facilitated.

Figure 3(a), 3(b), 2 shows the heatmap of the exact recovery rate of PCA, AN-RPCA, and M-MSR algorithm, respectively. It can be seen that the M-MSR algorithm can exactly recover the matrices when around 30% of the entries are severely corrupted. However, AN-RPCA algorithm can only recover matrices with around 20% corrupted entries and PCA can not recover the matrices with such severe corruptions for almost all dimensions and noise probabilities. Besides, we also compare the convergence time of the RPCA and M-MSR for exact recovery experiments. It can be observed that the running time of the M-MSR algorithm increases from 0.01s to 0.42s when the dimension of the matrices increases from $10 \times 10$ to $1000 \times 1000$, and the running time of the RPCA algorithm increases from 0.46s to 80.62s. The M-MSR algorithm is much more efficient than the RPCA algorithm, especially when applied on large datasets. This is an additional advantage of the M-MSR algorithm when dealing with rank-one matrix completion problems with corruptions on large datasets.

## G  Convergence Analysis for Arbitrary Graph

In this section, we provide the proof of Theorem 2.

### G.1  Proof of Theorem 2

*Proof. (sufficiency)* Suppose

$$u_i(t) = a_i k_i(t), \quad i \in [m],$$
$$v_j(t) = \frac{b_j}{k'_j(t)}, \quad j \in [n], \tag{18}$$

where $k_i(t)$ ($i \in [m]$) represents the value of the node in partition $V_u$ at iteration $t$, $k'_j(t)$ ($j \in [n]$) represents the value of the node in partition $V_v$ at iteration $t$, and the uncorrupted rank-one matrix is $ab^\top$. Let $M(t)$ and $m(t)$ be the maximum and minimum value of normal nodes at iteration $t$ respectively, i.e.,

$$m(t) \leq k_i(t) \leq M(t), \quad i \in [m] \cap \mathcal{N},$$
$$m(t) \leq k'_j(t) \leq M(t), \quad j \in [n] \cap \mathcal{N},$$

where $\mathcal{N}$ is the set of normal nodes. Our first step is to show that $M(t)$ and $m(t)$ are monotone bounded functions.

Let us consider a normal node $i \in [m]$. The value it receives from a neighbor $j$ at iteration $t + 1$ is $\frac{X_{ij}}{v_j(t)}$. If $j$ is also a normal node,

$$\frac{X_{ij}}{v_j(t)} = \frac{a_i b_j}{b_j / k'_j(t)} = a_i k'_j(t) \in [a_i m(t), a_i M(t)]. \tag{19}$$

On the other hand, if $j$ is corrupted, it is possible that $\frac{X_{ij}}{v_j(t)}$ is not in the interval $[a_i m(t), a_i M(t)]$. However, $\mathcal{G}(\Omega)$ is a $F$-local nodes-corrupted graph; and the largest and smallest $F$ values of $\frac{X_{ij}}{v_j(t)}$

are removed when updating $u_i(t+1)$. In other words, after filtering, the values the node $i$ receives from its neighbors are in the interval $[a_i m(t), a_i M(t)]$. Because $u_i(t+1)$ is a convex combination of such filtered values, we have

$$u_i^{(t+1)} \in [a_i m(t), a_i M(t)].$$

which implies

$$m(t) \le k_i(t+1) \le M(t).$$

We next make a similar argument for a normal node $j \in [n]$. The value $j$ receives from a neighbor $i$ at iteration $t+1$ is $\frac{X_{ij}}{u_i(t+1)}$. If $i$ is also normal node, then

$$\frac{X_{ij}}{u_i(t+1)} = \frac{a_i b_j}{a_i k_i(t+1)} = \frac{b_j}{k_i(t+1)} \in \left[ \frac{b_j}{M(t)}, \frac{b_j}{m(t)} \right]. \tag{20}$$

On the other hand, if $i$ is corrupted, it is possible that $\frac{X_{ij}}{u_i(t+1)}$ is not in the interval $[\frac{b_j}{M(t)}, \frac{b_j}{m(t)}]$. However, when we update $v_j(t+1)$, the largest and smallest $F$ values of $\frac{X_{ij}}{u_i(t+1)}$ are also removed. As a result, $v_j^{(t+1)} \in [\frac{b_j}{M(t)}, \frac{b_j}{m(t)}]$, which implies

$$m(t) \le k_j'(t+1) \le M(t).$$

We have thus derived that $M(t+1) \le M(t)$, $m(t+1) \ge m(t)$, i.e., $M(t)$ and $m(t)$ are both monotone bounded functions. Recall the property of skew-nonamplifying in (9) and (10), this also implies that M-MSR algorithm is skew-nonamplifying.

Next, that $M(t)$ and $m(t)$ are monotone bounded functions means each of them has some limits. Suppose the limit of $m(t)$ is $k_m$, the limit of $M(t)$ is $k_M$. If we have $k_M = k_m = k$, where $k$ is a positive constant, then all the normal nodes will asymptotically converge to $k$ at sometime $T$ and we can get

$$\boldsymbol{u}(T)\boldsymbol{v}(T)^\top = \begin{bmatrix} a_1 k_1(T) \\ a_2 k_2(T) \\ \cdots \\ a_m k_m(T) \end{bmatrix} \cdot \begin{bmatrix} \frac{b_1}{k_1'(T)} & \frac{b_2}{k_2'(T)} & \cdots & \frac{b_n}{k_n'(T)} \end{bmatrix} = \boldsymbol{u}^* \boldsymbol{v}^{*\top}. \tag{21}$$

We will next prove (21) is actually always true.

Indeed, suppose $k_M \ne k_m$; then there exists some $\epsilon_0 > 0$, such that $k_M - \epsilon_0 > k_m + \epsilon_0$. Let $S_M(t, \epsilon)$ denote the set of normal nodes which have values greater than $k_M - \epsilon$ at time-setp $t$, and let $S_m(t, \epsilon)$ denote the set of normal nodes which have values smaller than $k_m + \epsilon$ at time-setp $t$. If we can find $\epsilon^* > 0, \bar{\epsilon}^* > 0, t^* < \infty$ so that $S_M(t^*, \epsilon^*)$ or $S_m(t^*, \bar{\epsilon}^*)$ is empty, then all the normal nodes have values strictly smaller than $k_M - \epsilon^*$ or strictly greater than $k_m + \bar{\epsilon}^*$. This would contradict the assertion that $k_M$ is the limit of $M(t)$, or contradicts the assumption that $k_m$ is the limit of $m(t)$, respectively. Thus our goal is to prove that such $\epsilon^*, \bar{\epsilon}^*, t^*$ do exist.

Let

$$0 < \epsilon < \frac{(\alpha/2)^{m+n-F}}{\frac{1-\alpha}{2-\alpha}(1 - (\alpha/2)^{m+n-F}) + (1 - \alpha + \sqrt{2})/\alpha} \epsilon_0. \tag{22}$$

Since we assumed $\mathcal{G}(\Omega)$ is $2F + 1$-robust, at the very least we have that any node in $V(\mathcal{G})$ has at least $2F + 1$ neighbors, so that

$$2F + 1 \le \min\{m, n\}.$$

Therefore, we have $m + n - F \ge 2$, and since $\alpha \le 1/2$, we have

$$0 < \epsilon < \epsilon_0.$$

Since we assumed that there exists $\epsilon_0 > 0$ such that $k_M - \epsilon_0 > k_m + \epsilon_0$. If we choose smaller value of $\epsilon_0$, the inequality $k_M - \epsilon_0 > k_m + \epsilon_0$ still holds. In this case, we can always choose $\epsilon_0$ be small enough such that

$$k_m \ge \frac{\alpha(1-\alpha)(\epsilon + \epsilon_0)^2 + \epsilon\epsilon_0}{\alpha\epsilon_0 - (1-\alpha)\epsilon}. \tag{23}$$

Choose $t_0$ so that $M(t_0) < k_M + \epsilon$, $m(t_0) > k_m - \epsilon$ (the existence of $t_0$ is guaranteed by the convergence of $M(t)$ and $m(t)$). Then consider the two disjoint subsets $S_M(t_0, \epsilon_0)$ and $S_m(t_0, \epsilon_0)$. If $S_M(t_0, \epsilon_0)$ or $S_m(t_0, \epsilon_0)$ is empty, we can directly let $\epsilon^* = \epsilon_0$, $\bar{\epsilon}^* = \epsilon_0$, $t^* = t_0$ and we are done. Therefore we just need consider the case that both $S_M(t_0, \epsilon_0)$ and $S_m(t_0, \epsilon_0)$ are nonempty. As $\mathcal{G}(\Omega)$ is $2F + 1$-robust, there exists a node $s$ in $S_M(t_0, \epsilon_0)$ or $S_m(t_0, \epsilon_0)$ such that it has at least $2F + 1$ neighbors outside.

Because both $S_M(t_0, \epsilon_0)$ and $S_m(t_0, \epsilon_0)$ consist of nodes from $V_u$ and $V_v$, which have different update rules, we need discuss the following four cases respectively.

**Case A:** If $s \in S_M(t_0, \epsilon_0)$ is a node in $V_u$, it has at least $2F + 1$ neighbors outside $S_M(t_0, \epsilon_0)$, out of which at least $F + 1$ must be normal. Therefore, after removing $F$ largest and $F$ smallest neighbors, $s$ still can receive values from at least one normal node outside of $S_M(t_0, \epsilon_0)$.

By the same argument made in Eq. (19), the values $s$ receives from any normal neighbor lies in the interval $[a_s m(t_0), a_s M(t_0)]$. Since $\mathcal{G}(\Omega)$ is a $F$-local corrupted graph and the largest and smallest $F$ values $s$ receives are removed, all the values $s$ receives from its neighbors lie in the interval $[a_s m(t_0), a_s M(t_0)]$, which implies $\forall j \in \Omega_s \backslash R_s(t_0)$,

$$\frac{X_{sj}}{v_j(t_0)} \leq a_s M(t_0).$$

Then according to the update rule of Eq. (6) we have

$$\begin{aligned}
u_s(t_0 + 1) &= \sum\nolimits_{j \in \Omega_s \backslash R_s(t_0)} w_{sj} \frac{X_{sj}}{v_j(t_0)} \\
&\leq (1 - \alpha) a_s M(t_0) + \alpha a_s (k_M - \epsilon_0) \\
&\leq (1 - \alpha) a_s (k_M + \epsilon) + \alpha a_s (k_M - \epsilon_0) \\
&\leq a_s [k_M - (\alpha \epsilon_0 - (1 - \alpha)\epsilon)],
\end{aligned}$$

which implies

$$\begin{aligned}
k_s(t_0 + 1) &\leq k_M - (\alpha \epsilon_0 - (1 - \alpha)\epsilon) \\
&= k_M - \epsilon_a,
\end{aligned} \tag{24}$$

where

$$\epsilon_a = \alpha \epsilon_0 - (1 - \alpha)\epsilon.$$

Since $0 < \alpha \leq \frac{1}{2}$, and $m + n - F \geq 2$, then

$$\epsilon < \frac{(\alpha/2)^{m+n-F}}{\frac{1-\alpha}{2-\alpha}(1 - (\alpha/2)^{m+n-F}) + (1 - \alpha + \sqrt{2})/\alpha} \epsilon_0 < \frac{\alpha}{1 - \alpha} \epsilon_0,$$

which implies

$$0 < \epsilon_a < \epsilon_0. \tag{25}$$

**Case B:** If $s \in S_M(t_0, \epsilon_0)$ is a node in $V_v$, then by the same argument, $s$ will receive a value from at least one normal node with value bounded above $k_M - \epsilon_0$.

Similarly to Eq. (20), the values $s$ receive from its normal neighbors lie in the interval $[\frac{b_j}{M(t_0)}, \frac{b_j}{m(t_0)}]$. Reprising the argument in Case 1, $\mathcal{G}(\Omega)$ is a $F$-local corrupted graph and the largest and smallest $F$ values $s$ receives are removed in each iteration. Thus according to the update rule (7) we have

$$\begin{aligned}
v_s(t_0 + 1) &= \sum\nolimits_{i \in \Omega'_s \backslash R'_s(t_0)} w'_{is} \frac{X_{is}}{u_i(t_0 + 1)} \\
&\geq (1 - \alpha) \frac{b_s}{M(t_0)} + \alpha \frac{b_s}{k_M - \epsilon_0} \\
&\geq (1 - \alpha) \frac{b_s}{k_M + \epsilon} + \alpha \frac{b_s}{k_M - \epsilon_0}.
\end{aligned}$$

Thus

$$k'_s(t_0 + 1) = \frac{b_s}{v_s(t_0 + 1)} \leq \frac{1}{(1-\alpha)/(k_M + \epsilon) + \alpha/(k_M - \epsilon_0)}$$

$$= \frac{k_M^2 + k_M\epsilon - k_M\epsilon_0 - \epsilon\epsilon_0}{k_M - (1-\alpha)\epsilon_0 + \alpha\epsilon}$$

$$= \frac{k_M(k_M - (1-\alpha)\epsilon_0 + \alpha\epsilon) - \alpha k_M\epsilon_0 + (1-\alpha)k_M\epsilon - \epsilon\epsilon_0}{k_M - (1-\alpha)\epsilon_0 + \alpha\epsilon}$$

$$= k_M - \frac{\epsilon\epsilon_0 + \alpha k_M\epsilon_0 - (1-\alpha)k_M\epsilon}{k_M - (1-\alpha)\epsilon_0 + \alpha\epsilon}.$$

Let

$$\epsilon_b = \frac{\epsilon\epsilon_0 + \alpha k_M\epsilon_0 - (1-\alpha)k_M\epsilon}{k_M - (1-\alpha)\epsilon_0 + \alpha\epsilon}.$$

Then,

$$\epsilon_b - \epsilon_a = \frac{\alpha(1-\alpha)(\epsilon + \epsilon_0)^2}{k_M - (1-\alpha)\epsilon_0 + \alpha\epsilon} > 0,$$

which based on the fact that $0 < \alpha < \frac{1}{2}$, and

$$k_M > k_m \geq \frac{\alpha(1-\alpha)(\epsilon + \epsilon_0)^2 + \epsilon\epsilon_0}{\alpha\epsilon_0 - (1-\alpha)\epsilon} > (1-\alpha)\epsilon_0 - \alpha\epsilon,$$

where the last inequality is true as

$$\frac{\alpha(1-\alpha)(\epsilon + \epsilon_0)^2 + \epsilon\epsilon_0}{\alpha\epsilon_0 - (1-\alpha)\epsilon} - [(1-\alpha)\epsilon_0 - \alpha\epsilon] = \frac{2\epsilon\epsilon_0}{\alpha\epsilon_0 - (1-\alpha)\epsilon} > 0,$$

where the denominator is positive as we have shown in (25). As a result, we have $\epsilon_b > \epsilon_a$ and

$$k'_s(t_0 + 1) \leq k_M - \epsilon_b < k_M - \epsilon_a. \tag{26}$$

**Case C:** If $s \in S_m(t_0, \epsilon_0)$ is a node in $V_u$, via a similar process, we can get

$$k_s(t_0 + 1) \geq k_m + \alpha\epsilon_0 - (1-\alpha)\epsilon = k_m + \epsilon_c. \tag{27}$$

**Case D:** If $s \in S_m(t_0, \epsilon_0)$ is a node in $V_v$, we can obtain

$$k'^{(t_0+1)}_s \geq k_m + \frac{\alpha k_m\epsilon_0 + \alpha k_m\epsilon - \epsilon k_m - \epsilon\epsilon_0}{k_m + (1-\alpha)\epsilon_0 - \alpha\epsilon}.$$

Let $\epsilon_d = \frac{\alpha k_m\epsilon_0 + \alpha k_m\epsilon - \epsilon k_m - \epsilon\epsilon_0}{k_m + (1-\alpha)\epsilon_0 - \alpha\epsilon}$, then

$$\epsilon_d - \epsilon_c = \frac{-\alpha(1-\alpha)(\epsilon + \epsilon_0)^2}{k_m + (1-\alpha)\epsilon_0 - \alpha\epsilon} < 0,$$

which implies $\epsilon_d < \epsilon_c$. However, according to (23), we can derive

$$\frac{\alpha(1-\alpha)(\epsilon + \epsilon_0)^2}{k_m + (1-\alpha)\epsilon_0 - \alpha\epsilon} \leq \frac{\alpha(1-\alpha)(\epsilon + \epsilon_0)^2}{\frac{\alpha(1-\alpha)(\epsilon+\epsilon_0)^2 + \epsilon\epsilon_0}{\alpha\epsilon_0 - (1-\alpha)\epsilon} + (1-\alpha)\epsilon_0 - \alpha\epsilon}$$

$$= \frac{\alpha(1-\alpha)(\epsilon + \epsilon_0)^2(\alpha\epsilon_0 - (1-\alpha)\epsilon)}{2\alpha(1-\alpha)(\epsilon + \epsilon_0)^2}$$

$$= \frac{1}{2}(\alpha\epsilon_0 - (1-\alpha)\epsilon) = \frac{1}{2}\epsilon_c,$$

which means $\epsilon_d \geq \frac{1}{2}\epsilon_c$, and

$$k'_s(t_0 + 1) \geq k_m + \epsilon_d \geq k_m + \frac{1}{2}\epsilon_c. \tag{28}$$

**Summary:** Let

$$\epsilon_1 = \epsilon_a = \alpha\epsilon_0 - (1-\alpha)\epsilon,$$

$$\bar{\epsilon}_1 = \frac{1}{2}\epsilon_c = \frac{1}{2}(\alpha\epsilon_0 - (1-\alpha)\epsilon),$$

we see that in each case, at least one normal node $s$ in $S_M(t_0, \epsilon_0)$ decreases to $k_M - \epsilon_1$ (or below), or one normal node in $S_m(t_0, \epsilon_0)$ increases to $k_m + \bar{\epsilon}_1$ (or above), or both. Therefore, if we define the sets $S_M(t_0 + 1, \epsilon_1)$ and $S_m(t_0 + 1, \bar{\epsilon}_1)$, then we either have

$$|S_M(t_0 + 1, \epsilon_1)| < |S_M(t_0, \epsilon_0)|,$$

or

$$|S_m(t_0 + 1, \bar{\epsilon}_1)| < |S_m(t_0, \epsilon_0)|,$$

or both. Since $\bar{\epsilon}_1 < \epsilon_1 < \epsilon_0$, we have

$$k_M - \epsilon_1 > k_M - \epsilon_0 > k_m + \epsilon_0 > k_m + \bar{\epsilon}_1,$$

which means $S_M(t_0 + 1, \epsilon_1) \subseteq S_M(t_0 + 1, \epsilon_0)$ and $S_m(t_0 + 1, \bar{\epsilon}_1) \subseteq S_m(t_0 + 1, \epsilon_0)$. As set $S_M(t_0 + 1, \epsilon_0)$ and set $S_m(t_0 + 1, \epsilon_0)$ are disjoint, set $S_M(t_0 + 1, \epsilon_1)$ and set $S_m(t_0 + 1, \epsilon_1)$ are disjoint too.

For $j \geq 2$, let

$$\epsilon_j = \alpha\epsilon_{j-1} - (1-\alpha)\epsilon,$$

$$\bar{\epsilon}_j = \frac{1}{2}(\alpha\bar{\epsilon}_{j-1} - (1-\alpha)\epsilon),$$

then $\epsilon_j < \epsilon_{j-1}$, $\bar{\epsilon}_j < \bar{\epsilon}_{j-1}$. If both sets $S_M(t_0 + j, \epsilon_j)$ and $S_m(t_0 + j, \bar{\epsilon}_j)$ are nonempty, we can repeat the analysis above for time-step $t_0 + j$. If we can still show

$$\text{Case A: } k_s(t_0 + j) \leq k_M - \epsilon_j, \tag{29}$$

$$\text{Case B: } k'_s(t_0 + j) \leq k_M - \epsilon_j, \tag{30}$$

$$\text{Case C: } k_s(t_0 + j) \geq k_m + \bar{\epsilon}_j, \tag{31}$$

$$\text{Case D: } k'_s(t_0 + j) \geq k_m + \bar{\epsilon}_j, \tag{32}$$

we can derive that either

$$|S_M(t_0 + j, \epsilon_j)| < |S_M(t_0 + j - 1, \epsilon_{j-1})|,$$

or

$$|S_m(t_0 + j, \bar{\epsilon}_j)| < |S_m(t_0 + j - 1, \bar{\epsilon}_{j-1})|,$$

or both. Since

$$|S_M(t_0, \epsilon_0)| + |S_m(t_0, \epsilon_0)| \leq |\mathcal{N}| = m + n - F,$$

then there exists $T \leq m + n - F$ such that at the end of the iteration $t_0 + T$, the set $S_M(t_0 + T, \epsilon_T)$ or set $S_m(t_0 + T, \bar{\epsilon}_T)$ will be empty or both. Moreover, if we can further show

$$\epsilon_T > 0, \quad \bar{\epsilon}_T > 0, \tag{33}$$

then we can conclude that $\epsilon_T$, $\bar{\epsilon}_T$, and $t_0 + T$ are exactly the $\epsilon^*$, $\bar{\epsilon}^*$, and $t^*$ we are looking for. Next we will show that the inequalities (29)–(33) are actually always true.

For $j = 2, \ldots, m + n - F$,

$$\begin{aligned}
\epsilon_j &= \alpha\epsilon_{j-1} - (1-\alpha)\epsilon \\
&= \alpha(\alpha\epsilon_{j-2} - (1-\alpha)\epsilon) - (1-\alpha)\epsilon \\
&= \alpha^j\epsilon_0 - (1-\alpha^j)\epsilon,
\end{aligned}$$

$$\begin{aligned}
\bar{\epsilon}_j &= \frac{1}{2}(\alpha\bar{\epsilon}_{j-1} - (1-\alpha)\epsilon) \\
&= (\frac{\alpha}{2})^j\epsilon_0 - \frac{1-\alpha}{2-\alpha}(1 - (\frac{\alpha}{2})^j)\epsilon,
\end{aligned}$$

then $\epsilon_j < \epsilon_{j-1}$, $\bar{\epsilon}_j < \bar{\epsilon}_{j-1}$, and $\bar{\epsilon}_j < \epsilon_j$ ($\bar{\epsilon}_1 = \frac{1}{2}\epsilon_1$). In other words, we have

$$\epsilon_j > \bar{\epsilon}_j \geq \bar{\epsilon}_{m+n-F} = (\frac{\alpha}{2})^{m+n-F}\epsilon_0 - \frac{1-\alpha}{2-\alpha}(1-(\frac{\alpha}{2})^{m+n-F})\epsilon > \frac{1-\alpha+\sqrt{2}}{\alpha}\epsilon > 0, \quad (34)$$

where the second inequality is obtained from assumption (22). As $T \leq m + n - F$, we can derive that inequalities (33) are always true.

Then we will prove the inequalities (29)–(32) when $j \geq 2$. For case A and case C, we can show that the inequalities (29), (31) are true by simply replace $\epsilon_0$ with $\epsilon_{j-1}$ in the analysis above. For Case B, we need further show

$$k_m \geq \frac{\alpha(1-\alpha)(\epsilon+\epsilon_{j-1})^2 + \epsilon\epsilon_{j-1}}{\alpha\epsilon_{j-1} - (1-\alpha)\epsilon}, \quad (35)$$

to prove inequality (30), and for Case D, we need further show

$$k_m \geq \frac{\alpha(1-\alpha)(\epsilon+\bar{\epsilon}_{j-1})^2 + \epsilon\bar{\epsilon}_{j-1}}{\alpha\bar{\epsilon}_{j-1} - (1-\alpha)\epsilon}, \quad (36)$$

to prove the inequality (32). To do this, consider function

$$f(x) = \frac{\alpha(1-\alpha)(\epsilon+x)^2 + \epsilon x}{\alpha x - (1-\alpha)\epsilon},$$

its derivative is

$$f'(x) = \frac{(1-\alpha)(\alpha^2 x^2 - 2\alpha(1-\alpha)\epsilon x + \epsilon^2(\alpha^2 - 2\alpha - 1))}{(\alpha x - (1-\alpha)\epsilon)^2}.$$

When $x > \frac{1-\alpha+\sqrt{2}}{\alpha}\epsilon$, we have $f'(x) > 0$, which means $f(x)$ is monotonically increasing in this interval. From equation (34), we have

$$\frac{1-\alpha+\sqrt{2}}{\alpha}\epsilon < \bar{\epsilon}_{j-1} < \epsilon_{j-1} < \epsilon_0,$$

where $j = 2, \ldots, m + n - F$. Therefore, for $j = 2, \ldots, m + n - F$, we have

$$k_m \geq \frac{\alpha(1-\alpha)(\epsilon+\epsilon_0)^2 + \epsilon\epsilon_0}{(\alpha\epsilon_0 - (1-\alpha)\epsilon)^2} = f(\epsilon_0) > f(\epsilon_{j-1}) > f(\bar{\epsilon}_{j-1}),$$

where the first inequality is the assumption (23). In this case, we complete the proof of inequality (35) and inequality (36).

Then in Case B, we can replace $\epsilon_0$ with $\epsilon_{j-1}$, and we have

$$k_M > k_m \geq \frac{\alpha(1-\alpha)(\epsilon+\epsilon_{j-1})^2 + \epsilon\epsilon_{j-1}}{\alpha\epsilon_{j-1} - (1-\alpha)\epsilon} > (1-\alpha)\epsilon_{j-1} - \alpha\epsilon.$$

Next by conducting the similar analysis as above, we can prove the inequality (30). In Case D, we can replace $\epsilon_0$ with $\bar{\epsilon}_{j-1}$, and we can derive

$$\frac{\alpha(1-\alpha)(\epsilon+\bar{\epsilon}_{j-1})^2}{k_m + (1-\alpha)\bar{\epsilon}_{j-1} - \alpha\epsilon} \leq \frac{1}{2}(\alpha\bar{\epsilon}_{j-1} - (1-\alpha)\epsilon),$$

based on inequality (36). We can also follow the similar analysis above to prove (32). Thus, we complete the proof of sufficiency.

(Necessity) To prove the necessity, we want to show if $\mathcal{G}(\Omega)$ is not $2F + 1$-robust, then there exists cases which can not achieve consensus by applying M-MSR algorithm. Since $\mathcal{G}(\Omega)$ is not $2F + 1$-robust, there exists a pair of nonempty and disjoint sets $S_1, S_2 \in V(\mathcal{G})$ such that each node in $S_1$ or $S_2$ has at most $2F$ neighbors outside. Suppose $S_1$ consists of normal nodes which have values $a$ (meaning $k_i = a$ or $k_i' = a$) while $S_2$ consists of normal nodes which have values $b$ (meaning $k_i = b$ or $k_i' = b$) with $a > b > 0$, let all the other normal nodes have the values inside the interval $(b, a)$ ($a < k_i < b$ or $a < k_i' < b$). Since $\mathcal{G}(\Omega)$ is $F$-local corrupted graph, we can let each normal node in $S_1$ have $F$ corrupted neighbors which always send the normal node with value corresponding to $k_i = a$ or $k_i' = a$. This is possible, since in our cases, the corrupted elements $X_{ij}$ can be any value. Also, let normal node in $S_2$ have $F$ corrupted neighbors which always send the normal node with value corresponding to $k_i = b$ or $k_i' = b$. Thus, for the normal nodes in $S_1$ and $S_2$, the values which are different from their own values will always be filtered and they can only use the values equal to their own values to update. In this case, the consensus can never be achieved.

# H  Convergence Analysis for Random Graphs

In this section, we present the proof of Theorem 1. To do that, we provide the sharp threshold of being $r$-connected, $r$-robust for $\mathcal{G}_{n,n,p}$ as well as some other related lemmas. Besides, we also present two lemmas about how to represent a $F$-local model as a $F$-total model.

We start our analysis from introducing some definitions for $\mathcal{G}_{n,n,p}$ which will be used in our proof.

**Definition 5.** $A(n) \approx B(n)$ *means* $A(n)/B(n) \to 1$ *as* $n \to \infty$.

**Definition 6.** *A **graph property** $\mathscr{P}$ is a class of graphs on vertex sets $L$ and $W$, which is closed under isomorphism. In particular, $|\mathscr{P}| \leq 2^{n^2}$.*

**Definition 7.** *A graph property $\mathscr{P}$ is **monotone increasing** if $\mathcal{G} \in \mathscr{P}$ implies $\mathcal{G} + e \in \mathscr{P}$, i.e., adding an edge $e$ to a graph $\mathcal{G}$ does not destroy the property.*

**Definition 8.** *Consider a function $p^*(n) = \frac{g(n)}{n}$, where $g(n) \to \infty$ as $n \to \infty$. Let $x$ be any function such that $x = o(g(n))$ and $x \to \infty$ as $n \to \infty$. Then $p^*(n)$ is a **sharp threshold** for a monotone increasing graph property $\mathscr{P}$ in the random graph $\mathcal{G}_{n,n,p}$ if*

$$\lim_{n \to \infty} \mathbb{P}(\mathcal{G}_{n,n,p} \in \mathscr{P}) = \begin{cases} 1 & p = (g(n) + x)/n \\ 0 & p = (g(n) - x)/n \end{cases}. \tag{37}$$

We start our analysis from a general lemma about the necessary condition such that any approach achieves consensus when there exists malicious nodes in a network.

**Definition 9.** *For an undirected arbitrary graph $\mathcal{G}$, let $c_p(\mathcal{G})$ denote the least number $k$ such by removing $k$ appropriately chosen vertices from $\mathcal{G}$ and the eges incident on then results in a graph that is not connected.*

**Definition 10.** [25] *Consider an undirected arbitrary graph $\mathcal{G}$, suppose each normal node begins with some private value $x_i(0) \in \mathbb{R}$ (The initial values can be arbitrary). The nodes interact synchronously by conveying their values to their neighbors in the graph. Each normal node updates its own value over time according to a prescribed rule, which is modeled as*

$$x_i(t+1) = f_i(x_j^i(\cdot)), j \in \Omega_i, i \in \mathcal{N},$$

*where $x_j^i(\cdot)$ is the value sent from node $j$ to node $i$ before time-step $t+1$. The update rule $f(\cdot)$ can be arbitrary deterministic function, and may be different for different nodes. Then the normal nodes of $\mathcal{G}$ are said to achieve **resilient asymptotic consensus** in the presence of malicious nodes if*

- *$\exists k \in \mathbb{R}$ such that $\lim_{t \to \infty} x_i(t) = k$ for all $i \in \mathcal{N}$,*
- *the normal values remains in the interval $[m(0), M(0)]$ for all $t$,*

*where $m(0)$, $M(0)$ are the initial values.*

**Lemma 1.** (Theorem 5.2 of [8], Proposition 6.2.2 of [14]) *Suppose there exists $F$ malicious nodes in an undirected arbitrary graph $\mathcal{G}$, the positions of the malicious nodes are unknown, then the necessary condition that $\mathcal{G}$ can achieve consensus on a fixed value regardless of the mechanism used is*

$$c_p(\mathcal{G}) \geq 2F + 1.$$

## H.1  $r$-connected for random bipartite graph

In this subsection, we provide the sharp threshold function of being $r$-connected for random bipartite graphs $\mathcal{G}_{n,n,p}$. The general outline of this proof is to show the sharp threshold function for graph property that minimum degree $\delta(\mathcal{G}_{n,n,p}) = r$ firstly, then show that the graph property for $\mathcal{G}_{n,n,p}$ being $r$-connected is equal to the graph property that $\delta(\mathcal{G}_{n,n,p}) = r$.

**Definition 11.** *For $\mathcal{G}_{n,n,p}$ and constant $r \in \mathbb{Z}_{\geq 1}$, let the properties of being $r$-**connected**, having **minimum degree** $\delta(\mathcal{G}_{n,n,p}) = r$ be denoted by $\mathscr{K}_r$, $\mathscr{D}_r$, respectively.*

Suppose $X_r$ is the random variable counting the number of the vertices with degree $r$ in $\mathcal{G}_{n,n,p}$, $\lambda_r(n)$ is the expectation of $X_r$, i.e., $\lambda_r(n) = \mathbb{E}(X_r)$. Let $\text{Po}(\lambda)$ be the Poisson distribution with parameter $\lambda$, i.e., $\mathbb{P}(X = k) = \frac{\lambda^k e^{-\lambda}}{k!}$. Let $\text{N}(0,1)$ be standard normal distribution.

(a) A 2-connected graph      (b) A 3-connected graph

Figure 7: Illustration example for Lemma 1. Red nodes represent malicious nodes, green nodes represent normal nodes. In a 2-connected graph, malicious node $b$ can prevent node $a$ from getting correct information from node $c$. In a 3-connected graph, there exists three disjoint paths from node $c$ to node $a$, hence it is possible to apply some strategy to eliminate the influence of malicious node $b$.

Firstly, we present the sharp threshold function for property $\mathscr{D}_r$, to do that, we introduce the following lemma from [22].

**Lemma 2.** [22] *If $np \to \infty$ but $np/n^\alpha = o(1)$ for every $\alpha > 0$, then the distribution of $X_r \to \mathrm{Po}(\lambda)$ if $\lambda_r(n) \to \lambda < \infty$ and if $\lambda_r(n) \to \infty$, then the distribution of $(X_r - \lambda_r(n))/\sqrt{\lambda_r(n)} \to \mathrm{N}(0,1)$.*

In the following lemma, we provide the threshold function for property $\mathscr{D}_r$.

**Lemma 3.** *Consider a random bipartite graph $\mathcal{G}_{n,n,p}$. For any constant $r \in \mathbb{Z}_{\geq 1}$,*

$$p^*(n) = \frac{\log n + (r-1)\log\log n}{n}$$

*is a sharp threshold function for the property $\mathscr{D}_r$.*

*Proof.* Let

$$p = \frac{\log n + (r-1)\log\log n + x}{n}, \tag{38}$$

where $x = o(\log\log n) \to \infty$ when $n \to \infty$. We will show that with this probability, we can obtain

(i) $\lambda_t(n) = \mathbb{E}(X_t) = o(1)$ when $t \leq r - 2$, $n \to \infty$.

(ii) $\lambda_{r-1}(n) = \mathbb{E}(X_{r-1}) = \frac{2e^{-x}}{(r-1)!}$ when $n \to \infty$.

(iii) $\lambda_r(n) = \mathbb{E}(X_r) \to \infty$ when $n \to \infty$.

To do this, we will consider the vertices with degree $t$ ($t \leq r - 1$) in $L$ and $W$ respectively, where the notation $L$ denotes the set of left-nodes of the bipartition of this graph $\mathcal{G}_{n,n,p}$, and $W$ denote the set of right nodes. Let

$$\mathbb{I}_v = \begin{cases} 1 & v \text{ is a vertex with degree } t \text{ in } \mathcal{G}_{n,n,p} \\ 0 & \text{otherwise} \end{cases},$$

then we can obtain

$$\mathbb{E}(\text{number of nodes with degree } t \text{ in } L)$$
$$= \mathbb{E}\left(\sum_{v \in L} \mathbb{I}_v\right)$$
$$= \sum_{v \in L} \mathbb{E}(\mathbb{I}_v)$$
$$= n\binom{n}{t}p^t(1-p)^{n-t}, \tag{39}$$

As

$$\binom{n}{t} = \frac{n(n-1)\cdots(n-t)}{t!} \approx \frac{n^t}{t!}, \tag{40}$$

$$p^t = \left(\frac{\log n + (r-1)\log\log n + x}{n}\right)^t \approx \left(\frac{\log n}{n}\right)^t, \tag{41}$$

$$(1-p)^{n-t} = e^{(n-t)\log(1-p)} = e^{-(n-t)\sum_{k=1}^{\infty}\frac{p^k}{k}}$$

$$= e^{-(n-t)p} \cdot e^{-(n-t)\sum_{k=2}^{\infty}\frac{p^k}{k}} = e^{-(n-t)p} \cdot e^{-o(1)} \approx \frac{e^{-x}}{n(\log n)^{r-1}}, \tag{42}$$

where $x = o(\log\log n) \to \infty$ when $n \to \infty$, and in (42), we applied

$$(n-t)\sum_{k=2}^{\infty}\frac{p^k}{k} < (n-t)\sum_{k=2}^{\infty}p^k = \frac{(n-t)p^2}{1-p} = o(1).$$

Plug (40), (41), (42) into (39), we can get

$$\mathbb{E}(\text{number of nodes with degree } t \text{ in } L\,) \approx n\frac{n^t}{t!}\left(\frac{\log n}{n}\right)^t \frac{e^{-x}}{n(\log n)^{r-1}} = \frac{e^{-x}}{t!}\frac{(\log n)^t}{(\log n)^{r-1}}.$$

Via similar process, we can obtain

$$\mathbb{E}(\text{number of nodes with degree } t \text{ in } W\,) \approx \frac{e^{-x}}{t!}\frac{(\log n)^t}{(\log n)^{r-1}}.$$

Then we have

$$\mathbb{E}(X_t) = \mathbb{E}(\text{number of nodes with degree } t \text{ in } L\,) + \mathbb{E}(\text{number of nodes with degree } t \text{ in } W\,)$$

$$\approx \frac{2e^{-x}}{t!}\frac{(\log n)^t}{(\log n)^{r-1}}. \tag{43}$$

Thus (i), (ii) and (iii) follows immediately.

For $t \leq r - 2$, observing that $X_t$ is a nonnegative random variable, and $E(X_t) \to o(1)$, we can derive that

$$\mathbb{P}(X_t \neq 0) \to o(1). \tag{44}$$

For $X_r$, as

$$np = \log n + (r-1)\log\log n + x \to \infty,$$

$$\frac{np}{n^{\alpha}} = \frac{\log n + (r-1)\log\log n + x}{n^{\alpha}} = o(1), \quad \forall \alpha > 0,$$

when $n \to \infty$. According to Lemma 2, we have

$$\frac{X_r - \lambda_r(n)}{\sqrt{\lambda_r(n)}} \to \mathrm{N}(0,1),$$

then we obtain

$$\mathbb{P}(X_r \neq 0) = 1 - \mathbb{P}(X_r = 0) \approx 1, \tag{45}$$

In summary, when $p$ has the value in (38), it is not likely that the nodes with degree up to $r - 2$ exist in $\mathcal{G}_{n,n,p}$. Meanwhile, the probability that the nodes with degree $r$ exist in $\mathcal{G}_{n,n,p}$ tends to be 1. Thus, the probability that minimum degree $\delta(\mathcal{G}_{n,n,p}) = r$ is decided by the existence of the nodes with degree $r - 1$. Consider (ii), since $x = o(\log\log n) \to \infty$ as $n \to \infty$, we have

$$\lambda_{r-1}(n) = \frac{2e^{-x}}{(r-1)!} = o(1),$$

then we can also obtain $\mathbb{P}(X_{r-1} \neq 0) \approx 0$. Thus we have

$$\mathbb{P}(\delta(\mathcal{G}_{n,n,p}) = r) \approx \mathbb{P}(X_{r-1} = 0) \approx 1.$$

If

$$p = \frac{\log n + (r-1)\log\log n - x}{n},$$
(46)

we can just replace $x$ with $-x$ in (43) and produce

$$\mathbb{E}(X_t) \approx \frac{2e^{-x}}{t!} \frac{(\log n)^t}{(\log n)^{r-1}}.$$

In this case, (i) and (iii) remain the same while (ii) is updated as: (ii) $\lambda_{r-1}(n) = \mathbb{E}(X_{r-1}) = \frac{2e^x}{(r-1)!} \to \infty$. Therefore, (44) and (45) still hold while

$$\frac{X_{r-1} - \lambda_{r-1}(n)}{\sqrt{\lambda_{r-1}(n)}} \to \mathrm{N}(0,1),$$

then we can derive that

$$\mathbb{P}(\delta(\mathcal{G}_{n,n,p}) = r) \approx \mathbb{P}(X_{r-1} = 0) \approx 0.$$

$\square$

Now, we are ready to present the sharp threshold function of being $r$-connected for $\mathcal{G}_{n,n,p}$.

**Lemma 4.** *Consider a random bipartite graph $\mathcal{G}_{n,n,p}$. For any constant $r \in \mathbb{Z}_{\geq 1}$,*

$$p^*(n) = \frac{\log n + (r-1)\log\log n}{n}$$
(47)

*is a sharp threshold function for the property $\mathscr{K}_r$.*

*Proof.* $\mathcal{G}_{n,n,p}$ is $r$-connected means by removing $r$ suitably chosen vertices (but not by removing less than $r$ vertices) $\mathcal{G}_{n,n,p}$ can be disconnected. Let this event be denoted by $\mathcal{A}(\mathcal{S}, \mathcal{T})$, where the removed vertices form set $S$, and $T$ is the smallest connected component of $\mathcal{G}_{n,n,p} \backslash S$. In this case, $T$ has no neighbor after removing $S$. Besides, every node in $S$ is incident with at least one edge leading to $T$, otherwise $\mathcal{G}_{n,n,p}$ can be disconnected be removing less than $r$ vertices. We want to show if

$$p = (1 + o(1))\frac{\log n}{n},$$
(48)

then

$$\mathbb{P}\left(\exists S, T, 2 \leq |T| \leq \frac{1}{2}(2n - r) : \mathcal{A}(S, T)\right) = o(1).$$
(49)

Here $T$ is upper bounded by $\frac{1}{2}(2n - r)$ as $T$ is assumed to be the smallest remaining component after removing $S$. We choose $p$ with the value in (48) as it includes all the values of $p'(t)$ and $p''(t)$, $t \in [r]$, where

$$p'(t) = \frac{\log n + (t-1)\log\log n + x}{n}$$
$$p''(t) = \frac{\log n + (t-1)\log\log n - x}{n}.$$

where $x = o(\log\log n) \to \infty$ when $n \to \infty$. According to the definition of the sharp threshold, $p'(t)$ and the $p''(t)$ are the probabilities we need discuss to show that $\frac{\log n + (t-1)\log\log n}{n}$ is the sharp threshold function of being $t$-connected. If (49) is true, the only case we need consider for event $\mathcal{A}(S, T)$ is $|T| = 1$. In other words, if $\mathcal{A}(S, T)$ happens, the remaining subgraph $\mathcal{G}_{n,n,p} \backslash S$ after removing $S$ from $\mathcal{G}_{n,n,p}$ consists of some isolated vertices and a huge component, where the isolated vertices have degree $r$. Therefore,

$$\mathbb{P}(G \in \mathscr{K}_r) \approx \mathbb{P}(G \in \mathscr{D}_r).$$

This is because $\delta(\mathcal{G}_{n,n,p}) = r$ means the connectivity of $\mathcal{G}_{n,n,p}$ is less than or equal to $r$. However, if the connectivity is less than $r$, according to (49), there exists some vertices have degree less than $r$, which contradicts $\delta(\mathcal{G}_{n,n,p}) = r$. On the other hand, (49) also implies that if $\mathcal{G}_{n,n,p}$ is $r$-connected,

then $\delta(\mathcal{G}_{n,n,p}) = r$. In this case, we can show that the sharp threshold of $\mathscr{K}_r$ is equal to the sharp threshold of $\mathscr{D}_r$, i.e., (47).

Next, we will prove (49) holds with $p = (1 + o(1))\frac{\log n}{n}$. Fix the set $S$ and $T$, where $S$ consists of exactly $s_1$ nodes from partition $L$ and exactly $s_2$ nodes from partition $W$, $T$ consists of exactly $t_1$ nodes $T$ from $L$, and exactly $t_2$ nodes from $W$. Under such assumptions, let $\mathbb{P}_{s_1,s_2,t_1,t_2}$ denote the probability that event $\mathcal{A}(S,T)$ happens. Let $\mathcal{A}_1$ denote the event that $T$ is connected, $\mathcal{A}_2$ denote the event that $T$ is not connected to any vertex in $\mathcal{G}_{n,n,p}\backslash(S \cup T)$, and $\mathcal{A}_3$ denote the event that each vertex in $S$ is incident with at least one edge leading to $T$. Event $\mathcal{A}(S,T)$ happens when event $\mathcal{A}_1$, $\mathcal{A}_2$, and $\mathcal{A}_3$ happen at the same time, thus we have

$$\mathbb{P}_{s_1,s_2,t_1,t_2} = \mathbb{P}(\mathcal{A}_1 \text{ and } \mathcal{A}_2 \text{ and } \mathcal{A}_3)$$
$$= \mathbb{P}(\mathcal{A}_1)\mathbb{P}(\mathcal{A}_2)\mathbb{P}(\mathcal{A}_3), \tag{50}$$

where in the second equality, we applied the fact that $\mathcal{A}_1$, $\mathcal{A}_2$, and $\mathcal{A}_3$ are independent of each other (the appearance of the edges in $\mathcal{G}_{n,n,p}$ are identical independent random variables, $\mathcal{A}_1$, $\mathcal{A}_2$, and $\mathcal{A}_3$ refers to different edges).

Next we consider the probability of event $\mathcal{A}_1$, $\mathcal{A}_2$, and $\mathcal{A}_3$ respectively. $T$ is a connected subgraph implies that it contains a spanning tree with $t_1 + t_2 - 1$ edges. According to [12], the number of different spanning trees in $K_{t_1,t_2}$ is $t_1^{t_2-1}t_2^{t_1-1}$. By applying the union bound, we have

$$\mathbb{P}(\mathcal{A}_1) \leq t_1^{t_2-1}t_2^{t_1-1}p^{t_1+t_2-1}.$$

The probability that $T$ is disconnected with $\mathcal{G}_{n,n,p}\backslash(S \cup T)$ is

$$\mathbb{P}(\mathcal{A}_2) = (1-p)^{t_1(n-s_2-t_2)+t_2(n-s_1-t_1)}.$$

As $\mathcal{G}_{n,n,p}$ is a bipartite graph, the vertices in $L$ can only be connected with vertices in $W$. Let $S_1$ be the subset of $S$ which contains all the nodes from $L$, and $S_2$ be the subset of $S$ which contains all the nodes from $W$. Similarly, suppose $T_1$ be the subset of $T$ which consists of all the nodes from $L$, and $T_2$ consists of all the nodes from $W$. Therefore, each vertex in $S$ is incident with at least one edge leading to $T$ implies that there exists at least $s_1$ edges between $S_1$ and $T_2$, and at least $s_2$ edges between $S_2$ and $T_1$, hence we have

$$\mathbb{P}(\mathcal{A}_3) \leq \binom{s_1 t_2}{s_1}p^{s_1}\binom{s_2 t_1}{s_2}p^{s_2}$$

Then we can bound $\mathbb{P}_{s_1,s_2,t_1,t_2}$ as following

$$\mathbb{P}_{s_1,s_2,t_1,t_2} = \mathbb{P}(\mathcal{A}_1)\mathbb{P}(\mathcal{A}_2)\mathbb{P}(\mathcal{A}_3)$$
$$\leq t_1^{t_2-1}t_2^{t_1-1}p^{t_1+t_2-1}(1-p)^{t_1(n-s_2-t_2)+t_2(n-s_1-t_1)}\binom{s_1 t_2}{s_1}p^{s_1}\binom{s_2 t_1}{s_2}p^{s_2}$$
$$\leq t_1^{t_2-1}t_2^{t_1-1}p^{t_1+t_2-1}e^{-p[t_1(n-s_2-t_2)+t_2(n-s_1-t_1)]}(t_2 ep)^{s_1}(t_1 ep)^{s_2},$$

where in the second inequality, we used the facts

$$\binom{n}{k} \leq \left(\frac{ne}{k}\right)^k, \quad (1-p) \leq e^{-p}, \quad \forall 0 \leq p \leq 1.$$

Now, by applying the union bound, we can bound the probability $\mathbb{P}(\exists S,T)$ in (49) as

$$\mathbb{P}(\exists S,T) \leq \sum_{s_1+s_2=r}\sum_{t_1+t_2=2}^{\frac{2n-r}{2}}\binom{n}{s_1}\binom{n}{s_2}\binom{n-s_1}{t_1}\binom{n-s_2}{t_2}\mathbb{P}_{s_1,s_2,t_1,t_2}$$

$$\leq \sum_{s_1+s_2=r}\sum_{t_1+t_2=2}^{\frac{2n-r}{2}}\left(\frac{ne}{s_1}\right)^{s_1}\left(\frac{ne}{s_2}\right)^{s_2}\left(\frac{(n-s_1)e}{t_1}\right)^{t_1}\left(\frac{(n-s_2)e}{t_2}\right)^{t_2}t_1^{t_2-1}t_2^{t_1-1}p^{t_1+t_2-1}$$
$$e^{-p[t_1(n-s_2-t_2)+t_2(n-s_1-t_1)]}(t_2 ep)^{s_1}(t_1 ep)^{s_2}$$

$$\leq \sum_{s_1+s_2=r}\sum_{t_1+t_2=2}^{\frac{2n-r}{2}}\left(net_2 epe^{pt_2}\right)^{s_1}\left(net_1 epe^{pt_1}\right)^{s_2}(enp)^{t_1+t_2}e^{-p[t_1(n-t_2)+t_2(n-t_1)]}p^{-1}$$

$$\leq \sum_{s_1+s_2=r}\sum_{t_1+t_2=2}^{\frac{2n-r}{2}}p^{-1}A^{s_1}B^{s_2}C, \tag{51}$$

where in the second inequality, we also applied $\binom{n}{k} \le \left(\frac{ne}{k}\right)^k$, in the third inequality, we applied

$$\left(\frac{t_1}{t_2}\right)^{t_2-t_1} \le 1, \quad (t_1 t_2)^{-1} \le 1$$

and in the last step,

$$A = e^2 n p t_2 e^{p t_2} = e^2 (1 + o(1)) t_2 n^{\frac{t_2 + o(t_2)}{n}} \log n,$$
$$B = e^2 n p t_1 e^{p t_1} = e^2 (1 + o(1)) t_1 n^{\frac{t_1 + o(t_1)}{n}} \log n,$$
$$C = (enp)^{t_1 + t_2} e^{-p[t_1(n-t_2) + t_2(n-t_1)]}.$$

Let $t_1 + t_2 = t$, we have

$$
\begin{aligned}
C &= (enp)^{t_1+t_2} e^{-p[t_1(n-t_2)+t_2(n-t_1)]} \\
&= (enp)^t e^{-npt} e^{2pt_1 t_2} \\
&\le (enp)^t e^{-npt + \frac{t^2 p}{2}} \\
&= \left(enpe^{-np + \frac{pt}{2}}\right)^t \\
&= D^t,
\end{aligned}
$$

where in the inequality, we used

$$2pt_1 t_2 = 2pt_1(t - t_1) \le \frac{t^2}{2} p,$$

as $f(x) = 2px(t - x)$ attains its maximum at $x = \frac{t}{2}$, and in the last step

$$D = enpe^{-np + \frac{pt}{2}} = e(1 + o(1)) n^{-1-o(1) + \frac{\frac{t}{2} + o(\frac{t}{2})}{n}} \log n.$$

Thus, we can further bound the probability $\mathbb{P}(\exists S, T)$ as

$$\mathbb{P}(\exists S, T) \le p^{-1} \sum_{s_1 + s_2 = r} \sum_{t=2}^{\frac{2n-r}{2}} A^{s_1} B^{s_2} D^t. \tag{52}$$

Since if $1 \le t_1 \le \log n$, $1 \le t_2 \le \log n$, then

$$A = O((\log n)^2), \quad B = O((\log n)^2), \quad D = n^{-1+o(1)},$$

if $t_1 > \log n$, $t_2 > \log n$, then

$$A = O(n^3), \quad B = O(n^3), \quad D \le n^{-\frac{1}{3}},$$

as $t < n$, and if $1 \le t_1 \le \log n$, $t_2 > \log n$, then

$$A = O(n^3), \quad B = O((\log n)^2), \quad D \le n^{-\frac{1}{3}},$$

if $1 \le t_2 \le \log n$, $t_1 > \log n$, then

$$A = O((\log n)^2), \quad B = O(n^3), \quad D \le n^{-\frac{1}{3}}.$$

No matter which case, we have

$$p^{-1} A^{s_1} B^{s_2} D^t = o(1),$$

Thus the sum in (52) is $o(1)$.

$\square$

We have found the threshold function of $r$-connected for $\mathcal{G}_{n,n,p}$, however, the sufficient and necessary conditions for M-MSR algorithm to succeed are defined with the property robustness. Therefore, we also need confirm the threshold function of $r$-robustness. The following lemma is an important step to derive this threshold function.

## H.2   $r$-robustness for random bipartite graph

**Definition 12.** *For $\mathcal{G}_{n,n,p}$ and constant $r \in \mathbb{Z}_{\geq 1}$, let $\mathscr{E}_r$ be the property that every subset of $V(\mathcal{G})$ with size up to $n$ is $r$-**reachable**.*

Here, $\mathcal{G}_{n,n,p}$ is a bipartite graph where the total number of the nodes is $2n$.

**Lemma 5.** *Consider random bipartite graph $\mathcal{G}_{n,n,p}$. Then*

$$\lim_{n\to\infty} \mathbb{P}(\mathcal{G}_{n,n,p} \in \mathscr{E}_r) = 1, \tag{53}$$

*if*

$$p(n) = \frac{\log n + (r-1)\log\log n + x}{n}, \tag{54}$$

*where $x = o(\log\log n)$ satisfying $x \to \infty$ when $n \to \infty$.*

*Proof.* Let $\mathcal{A}_e$ denote the event that there exists a subset of $V(\mathcal{G})$ with size less than $n$ is not $r$-reachable, then we have

$$\mathbb{P}(\mathcal{G}_{n,n,p} \in \mathscr{E}_r) = 1 - \mathbb{P}(\mathcal{A}_e).$$

To prove (53) holds, we can show that $\mathbb{P}(\mathcal{A}_e) = o(1)$ with probability $p$ in (54).

Recall that $L$ and $W$ are the vertex partitions of the bipartite graph $\mathcal{G}_{n,n,p}$. Consider a subgraph of $\mathcal{G}_{n,n,p}$ where there exists $k_1$ vertices from L and $k_2$ vertices from $W$. Denote this subgraph as $S$, and let the probability that $S$ is not $r$-reachable be $\mathbb{P}_{k_1,k_2}$. From the proof of Lemma 4, we know when $p$ has the value as in (54), the probability that a vertex has degree less than $r$ is $o(1)$. In other words, the probability that a subset consists of one node is not $r$-reachable is $o(1)$. Then by applying the union bound, we have

$$\mathbb{P}(\mathcal{A}_e) \leq \sum_{k_1+k_2=2}^{n} P_{k_1,k_2}.$$

Consider a vertex $j \in S$, $j$ is not $r$-reachable means it has less than $r$ neighbors from outside. If $j$ is a vertex in $L$, the probability that it is not $r$-reachable is

$$\sum_{i=0}^{r-1} \binom{n-k_2}{i} p^i (1-p)^{n-k_2-i}.$$

If $j$ is a vertex in $W$, the probability that it is not $r$-reachable is

$$\sum_{i=0}^{r-1} \binom{n-k_1}{i} p^i (1-p)^{n-k_1-i}.$$

As $S$ is not $r$-reachable implies that every vertex in $S$ is not $r$-reachable, also there exists $k_1$ vertices of $S$ in $L$ and $k_2$ vertices of $S$ in $W$. By applying the union bound, we have

$$\mathbb{P}_{k_1,k_2}$$

$$\leq \binom{n}{k_1}\binom{n}{k_2} \left(\sum_{i=0}^{r-1}\binom{n-k_2}{i}p^i(1-p)^{n-k_2-i}\right)^{k_1} \left(\sum_{i=0}^{r-1}\binom{n-k_1}{i}p^i(1-p)^{n-k_1-i}\right)^{k_2}$$

$$\leq \left(\frac{ne}{k_1}\sum_{i=0}^{r-1}n^i p^i(1-p)^{n-k_2-i}\right)^{k_1} \left(\frac{ne}{k_2}\sum_{i=0}^{r-1}n^i p^i\left(1-p\right)^{n-k_1-i}\right)^{k_2}$$

$$\leq \left(\frac{ne}{k_1}(1-p)^{n-k_2}r\left(\frac{np}{1-p}\right)^{r-1}\right)^{k_1} \left(\frac{ne}{k_2}(1-p)^{n-k_1}r\left(\frac{np}{1-p}\right)^{r-1}\right)^{k_2}$$

$$\leq \left(\frac{er}{(1-p)^{r-1}}\frac{n}{k_1}e^{-p(n-k_2)}(np)^{r-1}\right)^{k_1} \left(\frac{er}{(1-p)^{r-1}}\frac{n}{k_2}e^{-p(n-k_1)}(np)^{r-1}\right)^{k_2}, \tag{55}$$

where in the second inequality we applied the inequalities

$$\binom{n}{k} \leq \left(\frac{en}{k}\right)^k, \quad \binom{n-k}{i} \leq (n-k)^i \leq n^i,$$

and in the third inequality, we used

$$\sum_{i=0}^{r-1} \left(\frac{np}{1-p}\right)^i \leq r \left(\frac{np}{1-p}\right)^{r-1},$$

which based on the fact that $\frac{np}{1-p} > 1$, in the last inequality of (55), we applied $1 - p \leq e^{-p}$ for $0 \leq p \leq 1$.

Let $c_1$ be a constant satisfying $\frac{er}{(1-p)^{r-1}} \leq c_1$. For sufficiently large $n$, we have $0 < c_1 < 2er$. Let $k_1 + k_2 = k$, then (55) can be rewritten as

$$
\begin{aligned}
\mathbb{P}_{k_1,k_2} &\leq c_1^k \frac{n^k}{k_1^{k_1} k_2^{k_2}} e^{2k_1 k_2 p} e^{-pnk} (np)^{(r-1)k} \\
&\approx c_1^k \frac{n^k}{k_1^{k_1} k_2^{k_2}} e^{2k_1 k_2 p} \frac{e^{-kx}}{n^k (\log n)^{k(r-1)}} (\log n)^{k(r-1)} \\
&= c_1^k \frac{e^{2k_1 k_2 p}}{k_1^{k_1} k_2^{k_2}} e^{-kx},
\end{aligned}
\tag{56}
$$

where in the approximate equality, we applied

$$e^{-pnk} = e^{-k \log n - k(r-1) \log \log n - kx} = \frac{e^{-kx}}{n^k (\log n)^{k(r-1)}},$$

$$np = \log n + (r-1) \log \log n + x \approx \log n.$$

Now consider the term $e^{2k_1 k_2 p}$ and $k_1^{k_1} k_2^{k_2}$ in (56), which can be written as

$$e^{2k_1 k_2 p} = e^{2k_1(k-k_1)p},$$

$$k_1^{k_1} k_2^{k_2} = e^{k_1 \log k_1 + k_2 \log k_2} = e^{k_1 \log k_1 + (k-k_1) \log(k-k_1)}.$$

Let

$$f(x) = 2x(k-x)p,$$
$$g(x) = x \log x + (k-x) \log(k-x).$$

For $x \in [0, k]$, $f(x)$ attains its minimum at $x = \frac{k}{2}$, $g(x)$ attains its maximum at $x = \frac{k}{2}$, i.e.,

$$f(x) \leq f\left(\frac{k}{2}\right) = \frac{k^2 p}{2}, \quad g(x) \geq g\left(\frac{k}{2}\right) = k \log\left(\frac{k}{2}\right), \quad \forall x \in [0, k].$$

As a consequence, we have

$$e^{2k_1 k_2 p} \leq e^{\frac{k^2 p}{2}},$$

$$k_1^{k_1} k_2^{k_2} \geq e^{k \log\left(\frac{k}{2}\right)} = \left(\frac{k}{2}\right)^k,$$

and (56) can be bounded by

$$\mathbb{P}_{k_1,k_2} \leq c_1^k \frac{e^{\frac{1}{2}k^2 p}}{2^{-k} k^k} e^{-kx} = \left(2 c_1 e^{\frac{1}{2}kp - \log k} e^{-x}\right)^k.$$

Now we can bound the probability $\mathbb{P}(\mathcal{A}_e)$ as

$$\mathbb{P}(\mathcal{A}_e) \leq \sum_{k_1+k_2=2}^{n} \mathbb{P}_{k_1,k_2} = \sum_{k=2}^{n} \left(2 c_1 e^{\frac{1}{2}kp - \log k} e^{-x}\right)^k. \tag{57}$$

Consider the term $e^{\frac{1}{2}kp-\log k}$ in (57), let

$$f(k) = \frac{1}{2}kp - \log k.$$

Then

$$f'(k) = \frac{1}{2}p - \frac{1}{k} = \frac{1}{2}\frac{\log n}{n}(1 + o(1)) - \frac{1}{k}.$$

As k ranges in the interval $[2, n]$, $f'(k) = 0$ has only one solution in $[2, n]$, and $f'(2) < 0$ while $f'(n) > 0$, which implies

$$f(k) \leq \max\{f(2), f(n)\},$$

where

$$f(2) = 2p - \log 2 < 0,$$
$$f(n) = \frac{n}{2}\frac{\log n}{n}(1 + o(1)) - \log(n) < 0.$$

Therefore,

$$\mathbb{P}(\mathcal{A}_e) = \sum_{k=2}^{n}\left(2c_1 e^{\frac{1}{2}kp-\log k}e^{-x}\right)^k < \sum_{k=2}^{n}(2c_1 e^{-x})^k \leq \frac{4c_1 e^{-2x}}{1 - 2c_1 e^{-x}} = o(1),$$

where in the second inequality, we applied

$$2c_1 e^{-2x} \leq \frac{4er}{e^{2x}} < 1.$$

□

**Lemma 6.** *For any* $r \in \mathbb{Z}_{\geq 1}$*, if a graph* $\mathcal{G}$ *is* $r$*-robust, then* $\mathcal{G}$ *is at least* $r$*-connected.*

*Proof.* We will prove this lemma by contradiction. Suppose there exists a graph $G$ which is $r$-robust and its connectivity is less than or equal to $r - 1$. According to the definition of $r$-connected, there exists a subset of $V(G)$ with size $r - 1$ such that $G$ will be disconnected with the removal of this subset. In other words, if this specific subset is removed, there will be at least two components remains. Choose one of the remained components arbitrarily, let it be $S_1$, let the union of all the other remained components be $S_2$. Then $S_1$ and $S_2$ are nonempty and disjoint, however, none of the node in $S_1$ or $S_2$ has more than $r - 1$ neighbors outside, which means both $S_1$ and $S_2$ are not $r$-reachable. This contradicts our assumption that $G$ is $r$-robust (for every pair of disjoint and nonempty subsets of $V(G)$, at least one of them is $r$-reachable), thus we can prove that if $\mathcal{G}$ is $r$-robust, then $\mathcal{G}$ is at least $r$-connected.

**Definition 13.** *For* $\mathcal{G}_{n,n,p}$ *and constant* $\mathbb{Z}_{\geq 1}$*, let* $\mathscr{R}_r$ *be the property of* $r$***-robust***.

Now we are ready to present the sharp threshold for the property of $\mathscr{R}_r$. The following theorem is connected to the work [40] which analyzed the threshold of $2F + 1$ robustness in general random graphs.

**Theorem 3.** *Consider random bipartite graph* $\mathcal{G}_{n,n,p}$*. For any constant* $r \in \mathbb{Z}_{\geq 1}$*,*

$$p^*(n) = \frac{\log n + (r - 1)\log\log n}{n}$$

*is the sharp threshold function for property* $\mathscr{R}_r$*.*

*Proof.* Let

$$p = \frac{\log n + (r - 1)\log\log n + x}{n}, \tag{58}$$

where $x = o(\log\log n) \to \infty$ when $n \to \infty$. Recall the definition of $r$-robust, i.e., a graph $\mathcal{G}$ is $r$-robust if for every pair of nonempty, disjoint subsets of $V(\mathcal{G})$, at least one of the two sets is $r$-reachable. To show $\mathcal{G}_{n,n,p}$ is $r$-robust, consider any two disjoint and nonempty subsets of $V(\mathcal{G})$,

define the two sets as $S_1$ and $S_2$. Then at least one of the two sets has size up to $n$, without loss of generality, let this set be $S_1$. According to Lemma 5, with probability (58), we have

$$\lim_{n \to \infty} \mathbb{P}(S_1 \text{ is } r\text{-reachable}) = 1,$$

which implies

$$\lim_{n \to \infty} \mathbb{P}(\mathcal{G}_{n,n,p} \in \mathscr{R}_r) = 1.$$

Next consider $\mathcal{G}_{n,n,p}$ with

$$p(n) = \frac{\log n + (r-1)\log\log n - x}{n}, \tag{59}$$

where $x = o(\log\log n) \to \infty$ when $n \to \infty$. We want to show

$$\lim_{n \to \infty} \mathbb{P}(\mathcal{G}_{n,n,p} \in \mathscr{R}_r) = 0. \tag{60}$$

From Lemma 6, we know that if $\mathcal{G}_{n,n,p}$ is $r$-robust, then $\mathcal{G}_{n,n,p}$ is at least $r$-connected, which means if the connectivity of $\mathcal{G}_{n,n,p}$ is less than $r$, then $\mathcal{G}_{n,n,p}$ is not $r$-robust. Besides, according to Lemma 4, with probability (59) we have

$$\lim_{n \to \infty} \mathbb{P}(\mathcal{G}_{n,n,p} \in \mathscr{K}_t) = 0, \quad \forall t \geq r.$$

Thus, we can obtain (60). □

### H.3  Proof of Theorem 1

Now, we are ready to present the proof of Theorem 1.

*Proof.*

We first argue that any skew-nonamplifying matrix completion method in the $F$-local model can be used to achieve resilient consensus in the $F$-local model over the bipartite graph corresponding to the revealed entries (recall that the resilient consensus problem was defined earlier in Definition 11).

Indeed, to achieve consensus starting from the initial conditions $x_i(0)$, we simply reveal entries corresponding to the all-ones matrix, and initialize $u_i(0) = x_i(0)$ on the left-side of the bipartition and $v_i(0) = 1/x_i(0)$ on the right-hand side of the bipartition. We then apply the skew-nonamplifying matrix completion method.

We can then define the values $k_i(t), k_i'(t)$ of a node just as in Eq. (18). Then according to the skew-nonamplifying property, it follows that for normal nodes

$$k_i(t), k_j'(t) \in [\min_i x_i(0), \max_i x_i(0)]$$

Since

$$u_i(t)v_j(t) = k_i(t) \cdot \frac{1}{k_j'(t)} = \frac{k_i(t)}{k_j'(t)} \to 1,$$

for all the normal nodes, we obtain that the quantities $k_i(t), k_j(t)$ among the normal nodes achieve consensus, and, as already remarked above, the quantities always stay in $[\min_i x_i(0), \max_i x_i(0)]$. It follows that we have an algorithm for resilient consensus.

Next, according to Lemma 1, a necessary condition for resilient consensus in the $F$-local corrupted model is $c_p(\mathcal{G}) \geq 2F + 1$. It follows that a necessary condition for skew-nonamplifying matrix completion in the $F$ local model is $c_p(\mathcal{G}) \geq 2F + 1$.

But from the result of Lemma 4, we know that a sharp threshold for being $r$-connected is

$$p = \frac{\log n + (r-1)\log\log n}{n}, \tag{61}$$

where $x = o(\log\log n) \to \infty$, when $n \to \infty$. Therefore, if

$$p = \frac{\log n + 2F \log\log n - x}{n}, \tag{62}$$

then the graph $\mathcal{G}$ will be such that no skew-nonamplifying algorithm can guarantee convergence under the $F$-local model. This proves Theorem 1(b).

Next consider the sufficient condition to correctly recover $X$ by applying the M-MSR algorithm. According to Theorem 2, the sufficient condition that the normal rows and columns of $X$ can be correctly recovered by M-MSR is $\mathcal{G}(\Omega)$ is $2F + 1$-robust. On the other hand, Theorem 3 shows if $p$ has the value as in (61), $\mathcal{G}(\Omega)$ is $r$-robust. Thus, the sufficient condition that the normal rows and columns of $X$ can be correctly recovered is (62), proving Theorem 1(a). □

From Theorem 1 and Corollary 3, we can see when $\mathcal{G}(\Omega)$ is a random bipartite graph $\mathcal{G}_{n,n,p}$, the proposed M-MSR algorithm is the optimal algorithm in rank-one matrix completion problem with corruptions.

## H.4 Further Results

In practical applications, It is not easy to confirm if $\mathcal{G}(\Omega)$ is $F$-local nodes-corrupted model. However, in random graph, we can represent a $F$-local model with a $F$-total model, which is more convenient to be verified. The following lemma provides a bridge from $F$-total model to $F$-local model. Particularly, we provide the fraction of the corrupted nodes in $L$ and $W$, respectively, so that $\mathcal{G}(\Omega)$ can be $f$-fraction local model.

**Lemma 7.** *Consider a random bipartite graph $\mathcal{G}_{n,n,p}$, let $p \geq \frac{12(1+\eta)\log n}{n}$ for some $\eta > 0$, then corrupt $\alpha n$ left nodes and $\beta n$ right nodes uniformly at random, where $0 \leq \alpha, \beta < 1$. In this case, for each normal node, fraction of edges from every normal node leading to corrupted nodes is less than $f$ with high probability when $n \to \infty$ if*

$$\alpha \leq f - \epsilon_1, \tag{63}$$
$$\beta \leq f - \epsilon_2, \tag{64}$$

*where $0 < \epsilon_1, \epsilon_2 \leq f$ are any constants.*

*Proof.* First, we will show that the vertex degree of $\mathcal{G}_{n,n,p}$ is bounded with high probability when $n \to \infty$. Since the degree of each node in $\mathcal{G}_{n,n,p}$ is the sum of $n$ independent Bernoulli random variables with parameter $p$. Then for a node $i \in V(\mathcal{G})$, by applying Chernoff bound, we can obtain

$$\mathbb{P}(d_i - \mu \leq -\delta\mu) \leq e^{-\mu\delta^2/3},$$

where $d_i$ represents the degree of node $i$, $\delta$ is a constant satisfying $0 \leq \delta \leq 1$, $\mu$ is the expected degree of node $i$, *i.e.*, $\mu = np$. By applying the union bound, we have

$$\mathbb{P}(\delta(\mathcal{G}_{n,n,p}) \leq (1-\delta)np) \leq 2ne^{-np\delta^2/3} = 2e^{\log n - np\delta^2/3}, \tag{65}$$

where $\delta(\mathcal{G}(n,n,p))$ is the minimum degree of graph $\mathcal{G}(n,n,p)$. Let $\delta = \frac{1}{2}$, and apply the inequality $p \geq \frac{12(1+\eta)\log n}{n}$, we can rewrite (65) as

$$\mathbb{P}\left(\delta(\mathcal{G}_{n,n,p}) \leq \frac{1}{2}np\right) \leq 2e^{\log n - np\delta^2/3} \leq \frac{2}{n^\eta} = o(1),$$

which implies that

$$\mathbb{P}\left(\delta(\mathcal{G}(n,n,p)) > \frac{1}{2}np\right) = \mathbb{P}\left(\delta(\mathcal{G}(n,n,p)) > 6(1+\eta)\log n\right) \approx 1.$$

As $\delta(\mathcal{G}(n,n,p))$ is the minimum degree of the graph $\mathcal{G}(n,n,p)$, for every node $i$ in $\mathcal{G}(n,n,p)$, we also have

$$\mathbb{P}\left(d_i > 6(1+\eta)\log n\right) \approx 1. \tag{66}$$

Next, consider the number of corrupted neighbors for each normal node. Suppose $i \in L$, then the probability of one outgoing edge of $i$ leading to corrupted nodes is $\beta$. Hence, the expected number of corrupted neighbors for node $i$ is $\beta d_i$. Let $Y_j$ be a random variable which represents the expected number of corrupted neighbors of node $i$ given that the first $j$ edges leaving $i$ have been revealed. Then the sequence of random variables $Y_0, \cdots, Y_{d_i}$ is a martingale such that

$$|Y_{j+1} - Y_j| \leq 1, \quad \forall j \in [d_i].$$

Note that among the sequence of random variables, $Y_0$ is the expected number of the infected neighbors of node $i$, i.e., $Y_0 = \beta d_i$, $Y_{d_i}$ is the expected number of corrupted neighbors of node $i$ when all of the outgoing edges have been revealed, which is exactly the number of corrupted neighbors of node $i$. Thus, according to Azuma's Inequality, we have

$$\mathbb{P}\left(Y_{d_i} - Y_0 \geq \lambda d_i\right) \leq e^{-\frac{\lambda^2 d_i^2}{2 \sum_{j=1}^{d_i} 1^2}} = e^{-\frac{1}{2}\lambda^2 d_i}. \tag{67}$$

Let $\lambda = f - \beta$, according to (63), we have $\lambda \geq \epsilon_1$. By combining (66) and (67), we can derive that the following inequality holds with high probability

$$\mathbb{P}(Y_{d_i} - Y_0 \geq \lambda d_i) = \mathbb{P}(Y_{d_i} \geq f d_i) \leq e^{-\frac{1}{2}\lambda^2 d_i} < e^{-\frac{1}{2}\epsilon_1^2 6(1+\eta)\log n} = \frac{e^{3(1+\eta)\epsilon_1^2}}{n} = o(1),$$

which implies with high probability

$$\mathbb{P}(Y_{d_i} < f d_i) \approx 1.$$

Thus we complete the proof for the nodes in $L$, the proof for nodes in $W$ is similar. $\quad\square$

## I   Sign Determination

In this section, we present the details about the sign pattern determination for rank-one matrices.

In the crowdsourcing problem, we allow the existence of the adversaries, which can lead to some of the entries of $\hat{C}$ are corrupted. Besides, $\hat{C}$ is an empirical estimate for $ss^\top$. In other words, it is possible that we can not find a sign pattern for $s$ which perfectly matches with the sign pattern of $\hat{C}$. Therefore, our goal is to find a sign pattern for $s$ to minimize the number of mismatching elements of $\text{sign}(ss^\top)$ and $\text{sign}(\hat{C})$.

To start with, consider a two-coloring problem: given a graph, color every node with one of two colors(e.g., red or blue) minimizing the number of "violations", where we say a violation occurs for each edge connecting nodes of the same color. Next, we will transfer our sign pattern determination problem to a two-coloring problem. Suppose the nodes value of node $i$ is $|s_i|$, sign $+$ represent color blue, sign $-$ represent color red. If an edge has two same color incident nodes, we call this edge is a "same color" edge, otherwise is an "opposite" color edge. In our problem, we are given the pattern of the edges, and we aim to color the nodes. To be consistent with the two coloring problem, we introduce some new nodes and edges as following: if an edge is an "opposite color" edge, we will put a new node in the middle of this edge, then the original "opposite color" edge becomes two "same color" edges. The obtained new graph is denoted as $\tilde{G}$. Thus, if we can find a way to color $\tilde{G}$ so that it satisfies the two-coloring rule, we also can get the sign pattern we are looking for. In Figure 8, an example is provided to illustrate this process.

Figure 8: Illustration for the sign pattern determination with a two-coloring method

The next step is to solve the two-coloring problem. To do that, we define a stochastic matrix $A$ for graph $\tilde{G}$ as following: let $A_{ij} = \frac{1}{d_i}$, $A_{ji} = \frac{1}{d_j}$ whenever nodes $i$ and $j$ are connected, and $A_{ij} = A_{ji} = 0$ otherwise. Then compute the eigenvector $v$ of A corresponding to the smallest eigenvalue. Finally assign $+$ to node $i$ if $v_i > 0$ and $-$ to node $i$ if $v_i < 0$.

There are two reasons why this approach works for two-coloring problem. First, suppose there exists a perfect assignment solution for this issue. The the graph $\tilde{G}$ is a bipartite graph. In this case, according to [23], the smallest eigenvalue of $A$ will be $-1$ and the corresponding eigenvector will be

composed of $+1$s and $-1$s which corresponds to different components of the bipartite graph. Second, in the event that there is no perfect assignment, according to [23],

$$\lambda_r(n) = \min \sum_{E(\tilde{G})} 2x_i x_j, \tag{68}$$

$$\text{s.t.} \sum_{i=1}^{n} d_i x_i^2 = 1,$$

where $\lambda_r(n)$ is the smallest eigenvalue of $A$, $x_i$, $x_j$ are the elements of the eigenvector corresponding to $\lambda_r(n)$ which are connected by edge $(i,j)$. It can be seen the eigenvector $\boldsymbol{x}$ is the solution of the optimization problem (68) which minimizes the number of the same color edges of $\tilde{G}$.

## Footnotes

[4]Available at `https://sites.google.com/site/nlpannotations/`

[5]Available at `https://github.com/ipeirotis/Get-Another-Label/tree/master/data`