[Reviews · NeurIPS 2020]

Review 1

Summary and Contributions: This paper targets at the problem of reconstructing a partially-observed rank-one matrix when some of the observed entries are corrupted with arbitrary perturbations. A new algorithm is proposed to recover the original rank-one matrix with bounded recovery probability. Experiments on both synthetic and real datasets showed that the proposed method can achieve better performance than other state-of-the-art methods.

Strengths: The main contribution of this work is theoretical, i.e., the recovery probability bound in Theorem 1. To the best of my knowledge, the results are new to the problem and could be used for other similar problems, .e.g., low-rank matrix completion with corrupted entries.

Weaknesses: The bound of Theorem 1 is tight according to the authors. But the bound seems not very informative in general cases, i.e., the probability of exact recovery bound is usually very small. It may be better to have some assumptions about the adversaries so that a bound related to the adversarial behaviors may be derived for more practical scenarios. I am not sure how close are the current results of this paper to recommendation problem. The user-item rating matrix in recommender systems are usually low-rank but not rank-one. It is better to discuss more on this because the paper selected recommender system and collaborative filtering as its areas. ------------------------- I have read the authors' rebuttal and will keep my original scores for this paper.

Correctness: The theoretical and empirical results seem correct to me.

Clarity: The paper is well-written.

Relation to Prior Work: Prior works are clearly compared and discussed.

Reproducibility: Yes

Additional Feedback:


Review 2

Summary and Contributions: This paper studies the problem of reconstructing a rank-one matrix from observed entries in the presence of adversarial corruption. It proposes a robust version of the alternating minimization algorithm by removing the F largest and smallest values in the aggregation step. It shows that the proposed algorithm correctly recovers the underlying matrix if and only if the set of observed entries represented as a bipartite graph satisfies the so-called (2F+1)-robust condition. Further, it shows that the proposed algorithm is information-theoretically optimal when the set of observed entries is given by an ER random graph. These results are then applied to the problem of learning workers’ skills in crowd-sourcing under the David-Scene model. Extensive experimental results demonstrate the improvement of the proposed algorithm compared to the state-of-the-art in the adversarial scenario. --- Update based on response The authors addressed some of my concerns. I am satisfied with the response.

Strengths: The proposed algorithm combines the idea of alternating minimization and the robust aggregation in network consensus. The results and analysis are solid. The performance characterization of the proposed algorithm is tight and the optimality condition in the ER random graph case is sharp. The application to crowd-sourcing is important. Extensive numerical studies further justify the applicability of the obtained results.

Weaknesses: (1) The focus on rank-one setting is restrictive. It is unclear how the results in the paper can be extended to more general settings. (2) The idea of removing the F largest and smallest values and the so-called (2F+1)-robust condition are borrowed from the consensus literature. The convergence analysis also builds upon the previous work. These need to be discussed more thoroughly in the main paper with proper references/credits to previous work. (3) The baseline methods used in numerical comparisons are mostly not designed for adversarial settings. Comparisons to existing robust methods are needed to demonstrate the improvement of the proposed algorithm.

Correctness: Yes

Clarity: The writing is very well written and the results are clearly explained.

Relation to Prior Work: Yes. The paper has an adequate discussion of the prior work and how this work differs from prior work.

Reproducibility: Yes

Additional Feedback:


Review 3

Summary and Contributions: This paper has proposed a matrix completion method for the label inference problem in crowdsourcing under the classification (categorization) setting. This framework is particularly effective when the adversary workers, who intend to provide the incorrect labels, are involved. The writing is excellent and framework is straightforward to understand.

Strengths: - Very strong motivation and well-explained problem definition as well as background analysis. - The framework is elegant with nice theoretical ganrantees. - The experiments are extensive, many popular crowdsourcing data sets are covered.

Weaknesses: - One coid model assumption may not fit well with crowdsourcing tasks that have dramatically different item difficulties, which could be a potential issue. - A few key refererences that are highly related are not compared and explained. - Experiments needs further improvement to show the full effectiveness of the proposed framework.

Correctness: Technical details are mostly solid with minor typos without affecting the reading.

Clarity: Good writing. Clear and straightforward.

Relation to Prior Work: References need to be expanded and add a few baselines for experimental comprehensiveness.

Reproducibility: No

Additional Feedback: These are a few issues need to be addressed for acceptance: - The one-coin model is not that popular after the two-coin model's prevelance in 2012. I believe the authors may adopt his model for mathematical convinence, however, you should also add the popular two-coid models into comparison. These papers includes: 1. Variational Inference for Crowdsourcing. 2. Learning from the Wisdom of Crowds by Minimax Entropy. 3. Spectral Methods meet EM: A Provably Optimal Algorithm for Crowdsourcing 4. The KOS model also has a two-coin version. Please adopt it if possible. - The matrix completion methods are popular for crowdsourcing problems. The current state-of-the-art are tensor-based completions on workers' labeling tensors. Please compare with the following SOTA methods. 1. crowdsourcing via tensor augmentation and completion - The study of the adversary workers in crowdsourcing has been emerged in recent years, please elaborate the difference between the proposed framework and the exsiting works. 1. Avoiding Imposters and Delinquents: Adversarial Crowdsourcing and Peer Prediction. 2. MultiC2: An Optimization framework for learning from task and worker dual heterogeneity. - The experiments are mainly focus on the performance with adversary workers being added. However, in real applications, adversary could be easily eliminated by inserting gold standard questions into tasks. The workers with consistently wrong answers on these gold standard questions will be banned. Therefore, it is more important to test the accuracy when the adversary's portion among workers are very small. However, this work didn't show much improvement in such scenario. - The crowdsourcing data set is usually very sparse, I don't suggest to remove the workers who has less than 10 items labeled. With this modification, the inference problem will become extremely easy. So, what about the experimental results using the raw dataset without this strong preprocessing step? - Notation error: e.g., line 41, nuclear norm of matrix X, not matrix L. - Notation abuse: rank-one matrix M and M-classification tasks. ---- review after rebuttal ---- The authors have addressed most of my doubts, I'm glad that the two-coin model's results are added to increase the comprehensiveness of your framework. However, it is still bizarre that two-coin performs worse than one-coin. I will raise my score to 7 for a good response from the authors. Please add the explanation in the latest version to explain your new experimental results. I also agree with review #1 and have concerns on the rank-1 assumption which might be too strong? Unfortunately, it didn't receive an intuitive explanation. Please address your motivation for using rank-1 matrix completion, maybe build the connection (not only the difference) with the standard MF-based methods and nuclear norm minimization methods.


Review 4

Summary and Contributions: The authors studied the problem of matrix completion in the presence of corrupted entries. To address the problem, the authors proposed an algorithm, which integrates alternating minimization and extreme-value filtering to perform reliable matrix completion. Experimental results on 17 real data sets show the performance of the proposed method.

Strengths: (1) Some theoretical groundings are provided to support the validity of the proposed method. (2) Extensive experiments are conducted to demonstrate the performance of the algorithm.

Weaknesses: (1) The writing of the paper can be improved. Some notions are not well described. (2) It is unclear about the novelty and actual contribution for advancing the state of the art of the crowd sourcing. (3) Some key works are not considered in both literature review and comparison experiments.

Correctness: The claims and empirical methodology are clear and correct.

Clarity: No, The notations are messy. Many are used on-the-fly without clear definition.

Relation to Prior Work: The technical innovations are minor and they don't bring any conceptual breakthrough. They are just adhoc straightforward modifications of known algorithms.

Reproducibility: Yes

Additional Feedback: The approach and the solution to the problem are very well handled. However, it also suffers from multiple drawbacks. Below are the detailed comments: (1) The studied problem is not novel and the proposed technicals are somehow related to the existing works on matrix completion based crowdsourcing. (2) The notations are messy. Many are used on-the-fly without a clear definition. For example, in line 126~127, the authors introduce "W workers" and "M classes". Are they equivalent to "n" and "m" in Def. 1? If yes, the authors may want to clarify it and ensure the consistency through the paper. (3) According to Eq. 2, it is clear that D&S model does not consider the different capability/reliability of workers across different tasks, which could be critical in real applications. The authors may want to provide insightful discussion and empirical studies to show whether the M-MSR can handle the capability/reliability of workers and how in real scenarios. (4) The authors may want to conduct comparison experiments between the proposed one and two-coin model. It is interesting to know whether the proposed method can beat two coin model and to what extend. (5) Some related works are missing, e.g., Houping Xiao, Jing Gao, Zhaoran Wang, Shiyu Wang, Lu Su, Han Liu. A Truth Discovery Approach with Theoretical Guarantee. KDD'16 Jing Gao, Qi Li, Bo Zhao, Wei Fan, Jiawei Han: Mining Reliable Information from Passively and Actively Crowdsourced Data. KDD 2016: 2121-2122 Houping Xiao, Jing Gao, Qi Li, Fenglong Ma, Lu Su, Yunlong Feng, Aidong Zhang: Towards Confidence Interval Estimation in Truth Discovery. IEEE Trans. Knowl. Data Eng. 31(3): 575-588 (2019) ########################################################## Thank you for the feedback. Some of my concerns have been addressed, and I've adjusted my score accordingly. I've also included additional references.

[Author Response · NeurIPS 2020]

We want to thank the reviewers for their careful reading of the paper and their many constructive comments. Due to
space constraints, we are not able to reply to every point made by the reviewers and focus our response on the most
important points. Naturally, all of our responses here will be incorporated into the final version of the paper.
**Additional baselines and the two-coin model:** We would like to thank Reviewer 3 for suggesting several highly
relevant references with additional algorithms. We therefore compared the M-MSR method with 8 new baseline
methods, where 6 of them are two-coin methods (Minmax[1], Entropy(O)[2], EM-twocoin[3], BP-twocoin[3], MFA-twocoin[3],
and M-MSR-twocoin) and 2 of them are tensor methods (EM-MV[4], EM-SM[4]).

Additionally, all reviewers suggest we should test our method on the two-coin model: here the labels are binary and the
probability of giving the correct answer depends on the label, so that each worker is parametrized by two parameters.
Besides the above methods and M-MSR, we will also simulate a natural extension of M-MSR to the rank-2 case:

$$x_i^{t+1} = \arg\min_x \sum_{j \,|\, (i,j)\in\Omega} (x^T y_j - C_{ij})^2; \qquad y_j^{t+1} = \arg\min_y \sum_{i \,|\, (i,j)\in\Omega} (x_i^T y - C_{ij})^2,$$

where $x_i \in \mathbb{R}^2$ denotes the $i$th row of $X$ and $y_j \in \mathbb{R}^2$ denotes the $j$th row of $Y$; to deal with the corrupted entries, we
throw out the largest and smallest $F$ values among the quantities $|C_{ij}| / \|y_j\|_2$ inside the minimization problem in each
step. We call this algorithm the M-MSR-twocoin algorithm. It is not hard to see that, if the true proportion of "0" and
"1" labels in the data-set is known (more on this later), skill estimation in the two-coin can be reduced to a rank-2 matrix
recovery problem to which this algorithm can be applied.

We had time to implement the same experiments as in the paper on 5 real datasets and one synthetic dataset. The
synthetic dataset is created following the two-coin model. The results are shown in Figure 1. *We can see in the figure*
*that M-MSR outperforms all the baseline methods on all 6 datasets. Surprisingly, M-MSR even beats its own natural*
*generalization to rank-2, even though this generalization has the implicit advantage of knowing the true proportion of*
*"0" and "1" answers; and it can also beat all the methods tailored to the two-coin model on the synthetic data set once*
*adversaries are introduced.*

We have spent substantial efforts during the rebuttal period implementing these new baselines and we believe the
results considerably strengthen the case for the effectiveness of M-MSR. While it may not be surprising that M-MSR
outperforms various two-coin methods ( since M-MSR is designed with adversaries in mind) it is quite surprising that
M-MSR outperforms its rank-2 generalization.
**Reviewer 2:** **More discussion for the consensus literature.** Indeed, the reviewer is correct. We will move some of
Section 5.1 to the supplementary information which will give us the space to discuss this connection.
**Reviewer 3:Reproducibility.** We will publish all the code with the final version paper on github.
**Differences between our work and literature[5], literature[6].** One difference from literature[5] is that we aim to label
the class of the tasks, and they aim to rate the tasks (not necessarily integer) and find the ones with high quality. The
main difference, however, is that they require the ratings of a small portion of tasks are known. Literature[6] and our work
both study the issue of worker ability estimation, however, we mainly focus on the adversarial setting while they do not.
**Use of "gold standard" questions.** One advantage of M-MSR is that it can tolerate arbitrary adversaries, including
adversaries that return large fraction of correct answers (this can be be seen in Figure 1(g) in the paper), which means
M-MSR can deal even with smart adversaries that manage to pass the gold question test.
**Experimental results using raw datasets.** Among the 17 real datasets we used, 11 of them had each worker with 10
tasks or more, so no removal of workers was necessary. For the other 6 datasets, we implemented experiments on 4 of
them (Dog, Adult, Fashion1, and Fashion2) using raw data without removals; this did not change the results. The results
in Figure 1 below also use raw data without removals. Experiments on remaining 2 raw datasets (Web and TREC) will
be added in final version of the paper (Web is a multiclass data set and some of the two-coin methods cannot deal with
that; TREC is very large and some of the methods take a very long time to run on it).
**Reviewer 4:** **Notation.** Note that we consider two problems: recovery of an $m \times n$ rank-1 matrix and crowdsourcing
with $W$ workers and $M$ classes. We reduce the crowdsourcing problem to rank-1 recovery of a $W \times W$ matrix.

**Figure 1:** Experimental results. Solid lines are the average of 20 runs, and the shaded region show the standard deviation. The total number of workers of the datasets: Adult 269, Dog 109, Fashion1 196, Fashion2 198, Temp 76, Synthetic two-coin 40. Adult and Dog are multiclass datasets, hence the methods which are designed only for binary datasets are not shown. The synthetic dataset is created following two-coin model rule, where the worker probabilities of giving correct answers are uniform in $[0.5, 1]$ for both classes, and the true answers have equal probabilities of coming from each class.

[1] Zhou et al. Learning from the Wisdom of Crowds... [2] Zhou et al. Aggregating Ordinal Labels... [3] Liu et al. Variational Inference for Crowdsourcing [4] Zhang et al. Spectral Methods meet EM... [5] Steinhardt et al. Avoiding Imposters... [6] Zhou et al. MultiC$^2$: An Optimization framework...


[Meta-Review · NeurIPS 2020]

The paper addresses the problem of rank one matrix completion when some rows can be adversarially corrupted. Motivations include crowdsourcing under the Dawid-Skene model and matrix completion in Yelp and other online platform. The reviewers agree that the topic and results are of interest to the community and recommend acceptance. There were three themes of concerns that were brought up in multiple reviews, as well as in my own opinion when reading the paper: (1) the assumptions underlying the work, and (2) appropriate credit and references to prior work. A reviewer also brought up concerns with (3) results. I discuss these issues in more detail below. IMPORTANT: In the revised version, please issue appropriate clarifications, references, and discussion to address these issues brought up by reviewers. (1) Assumptions underlying the work: Multiple reviewers question the application to recommendation systems "The user-item rating matrix in recommender systems are usually low-rank but not rank-one." "The focus on rank-one setting is restrictive. It is unclear how the results in the paper can be extended to more general settings." "I also agree with review #1 and have concerns on the rank-1 assumption which might be too strong? Unfortunately, it didn't receive an intuitive explanation." This is indeed a significantly restrictive assumption which was brought up by multiple reviewers but was not addressed in the rebuttal. The paper however seems to overclaim application to this setting. In my opinion, connections to this setting should be discussed with a clear highlight on the disconnect, and are best left as future work. Reviewer 3 also complains "One coin model assumption may not fit well with crowdsourcing tasks that have dramatically different item difficulties, which could be a potential issue." I do agree that this is an extremely stylized model and the community has moved beyond the one coin as well as the two coin model since neither of these models capture tasks that have dramatically different item difficulties. See instead the papers https://arxiv.org/pdf/1602.03481.pdf and  http://arxiv.org/pdf/1606.09632. In order to give a complete picture to the reader, please include a brief discussion on one coin, two coin, and more general models, possibly mentioning extensions to other models as future work. (2) Appropriate credit and references to prior work: A reviewer criticizes "The idea of removing the F largest and smallest values and the so-called (2F+1)-robust condition are borrowed from the consensus literature. The convergence analysis also builds upon the previous work." These need to be discussed more thoroughly in the main paper with proper references/credits to previous work. Reviewers 2, 3 and 4 provide relevant literature in crowdsourcing. (3) Results Reviewer 1 is also concerned about Theorem 1 being uninformative. In my own reading, I could not find the definition of "with high probability" anywhere. Is it 1-e^-n? or 1-1/loglog(n)? Or something else? Please issue appropriate clarifications regarding the two aforementioned points. In my reading, there is another issue that should be clarified in the camera ready. In the experiments with real data, the leftmost points are those which correspond to real data. The entire remaining plot is semi-synthetic. But if we look at what happens only on the real data, the algorithm performs comparably or frequently much worse than prior work. We do not know how many workers in this real data were adversarial. If there were many, then the algorithm is performing well only when a lot more adversarial workers are added in the right points. If there weren't then it goes against the crowdsourcing motivation of this work. The performance of the proposed algorithm improves as more adversaries are synthetically added. Such a tradeoff (between algorithms robust to adversaries vs. not, and where there exist adversaries vs. not) is natural, and must at least be discussed, also making it clear that the proposed algorithm does perform worse on the actual data obtained from the real world. --- Note: Review 4 was inappropriate in criticisms of the form "does not compare with related work" without actually providing concrete instances of what was not compared with. I've downweighted these parts of review 4 and requested the reviewer to update their review accordingly.